# Linear and Nonlinear Optical Properties of Quadrupolar Bithiophenes and Cyclopentadithiophenes as Fluorescent Oxygen Photosensitizers

Nicolas Richy [1], Safa Gam [1,2,3], Sabri Messaoudi [2,3,4], Amédée Triadon [1,5], Olivier Mongin [1], Mireille Blanchard-Desce [6], Camille Latouche [7], Mark G. Humphrey [5], Abdou Boucekkine [1,*], Jean-François Halet [8,*] and Frédéric Paul [1,*]

[1] Univ Rennes, CNRS, Institut des Sciences Chimiques de Rennes (ISCR)—UMR 6226, F-35000 Rennes, France
[2] Faculty of Sciences of Bizerte FSB, University of Carthage, Jarzouna 7021, Tunisia
[3] Laboratory of Materials, Molecules and Applications, IPEST, University of Carthage, SidiBou Said Road, B.P. 51, La Marsa 2070, Tunisia
[4] Department of Chemistry, College of Science, Qassim University, Buraidah 51452, Saudi Arabia
[5] Research School of Chemistry, Australian National University, Canberra, ACT 2601, Australia
[6] Univ. Bordeaux, CNRS, Bordeaux INP, ISM (UMR5255), Bat A12, 351 Cours de la Libération, F-33405 Talence, France
[7] Nantes Université, CNRS, IMN (Institut des Matériaux de Nantes Jean Rouxel), F-44000 Nantes, France
[8] CNRS—Saint-Gobain—NIMS, IRL 3629, Laboratory for Innovative Key Materials and Structures (LINK), National Institute for Materials Science (NIMS), Tsukuba 305-0044, Japan
* Correspondence: abdou.boucekkine@univ-rennes1.fr (A.B.); jean-francois.halet@univ-rennes1.fr (J.-F.H.); frederic.paul@univ-rennes1.fr (F.P.)

**Abstract:** The linear and nonlinear optical properties of two quadrupolar bithiophenes and two quadrupolar cyclopentadithiophenes have been investigated. At the 5,5′ positions of the central bi/dithiophene units, the molecules possess 1,4-phenylalkynyl groups that bear either electron-donating ($NPh_2$) or electron-withdrawing ($SO_2CF_3$) groups. The optical properties were experimentally studied and modelled via quantum chemistry computations of key configurations and conformations. All the compounds show good light harvesting efficiency due to their strong absorption in the visible range. These fluorescent compounds are also good two-photon absorbers in the NIR range that can photosensitize oxygen in toluene. DFT calculations reveal that the mixtures of conformers in a solution show similar linear optical properties. TD-DFT calculations reproduce the experimental spectroscopic data fairly well, including vibronic couplings in the fluorescence spectra. The lowest excited state for two-photon absorption corresponds to the $S_2$ state. The roles of the $SO_2CF_3$ and $NPh_2$ terminal groups on the nonlinear response were analyzed for possible bio-oriented applications, with the cyclopentadithiophenes showing the most promising figures of merit.

**Keywords:** bithiophenes; DFT calculations; fluorescence; oxygen photosensitization; two-photon absorption

## 1. Introduction

The pursuit of dyes exhibiting two-photon absorption (2PA) at wavelengths located in the first or second biological window has attracted tremendous research effort due to the inherent advantages of 2PA excitation over the more conventional one-photon (1PA) excitation [1–3], turbo-charging the search for fluorophores exhibiting larger and larger two-photon absorption cross-sections [4]. The replacement of 1,4-phenylene rings by less aromatic and more polarizable 2,5-thienylene groups is a trick often used by synthetic chemists to enhance the nonlinear optical properties of (poly)cyclic organic molecules [5–14]. Poly(2,5-thienyl) moieties are effective two-photon absorbers in their own right [15,16], with their incorporation boosting 2PA in various molecular architectures, especially when a specific multipolar symmetry is preserved [17]. "Extended" bithiophenes and polythiophenes

were recently demonstrated to combine enhanced 2PA responses and appropriate emissive properties for biological imaging, cellular imaging and even DNA/RNA sensing [18–21]. The 2PA properties of dipolar push–pull 1,4-(bis(2,5-thienyl)arylene chromophores bearing electron-donating (D) and -withdrawing (A) termini have also been investigated for similar uses [22].

The fine-tuning and optimization of compounds for a given task in biophotonics require a deep understanding of the influence of structural variations on the optical properties of interest [23]. In that respect, the study of bithiophene derivatives such as **1a,b** (Scheme 1) may clarify the role of endgroups (electron-releasing vs. electron-withdrawing), while rigid, non-centrosymmetric analogues, such as the 4,4-dibutyl-4*H*-cyclopenta [2,1-b:3,4-b']dithiophene **2a,b**, may shed light on the importance of symmetry and/or conformational flexibility along the long axis of these molecules and provide a clearer picture of the interplay between the electronic structure and optical properties. Comparison with the known fluorene homologues **3a,b'** [24,25], which are also strongly emissive two-photon absorbers, would enhance this study by clarifying the impact of the 1,4-phenylene replacement by 2,5-thienyl units in these molecular scaffolds. In addition to experimental determination of the properties of interest, crucial information on these phenomena can often simply be gained by performing quantum chemical computations on a library of carefully selected models, since density functional theory (DFT) and its time-dependent extension (TD-DFT) are now accurate enough to rationalize both the linear and nonlinear optical properties of small molecules [14,26]. Such calculations are also routinely used to determine the key excited states in these processes [27].

**Scheme 1.** Bithiophene-cored quadrupoles **1a,b**, their cyclopentadithiophene-cored analogues **2a,b** and some known fluorenyl-based references **3a,b** and **3b'** (Bu = *n*-butyl, Non = *n*-nonyl).

Despite the simplicity of quadrupolar model compounds such as **1a,b** and **2a,b**, they have never been experimentally studied (to the best of our knowledge), while only **2a** was briefly modelled by DFT and compared to **3a** at the same level of theory [28]. According to these theoretical results, the cyclopentadithiophene derivative **2a** should be a better two-photon absorber than **3a**, pointing toward an appealing potential for cyclopentadithiophenes as a promising class of two-photon fluorescent dyes. In order to confirm the merit of these compounds relative to their fluorene homologues **3a,b** and to assess their potential applications in biophotonics in a more definitive way, we have synthesized **1a,b** and **2a,b** and studied their photophysical properties. We herein report (i) their synthesis and characterization and (ii) the experimental determination and computational modelling of their

one- and two-photon absorption, emission and oxygen photosensitization behavior before (iii) briefly discussing their potential for bio-oriented uses, as well as (iv) the role played by the sulfur atoms on the optical properties of interest. The data presently obtained for **1a,b** and **2a,b** are analyzed and contrasted with the experimental [24,25] and computational [28] data available for **3a,b** and **3b′**.

## 2. Results

### 2.1. Syntheses of the New Derivatives

*2,2′-Bithiophene Derivatives.* Double Sonogashira coupling of commercially available 5,5′-dibromo-2,2′-bithiophene (**4**) with alkyne **5** [29], followed by desilylation in situ with TBAF afforded the push–push quadrupole **1a** in 33% yield (Scheme 2). The dibromo precursor **4** was also converted into the dialkyne **6b** [30] in a two-step sequence involving a reaction with trimethylsilylacetylene, yielding **6a** as an intermediate and then **6b** after removal of the TMS groups. The pull–pull chromophore **1b** was then obtained in a 30% yield by coupling **6b** with the bromoarene **7** [25,31].

**Scheme 2.** Synthesis of **1a,b**. Reagents and conditions: (a) TBAF, Pd(PPh$_3$)$_2$Cl$_2$, CuI, toluene/NEt$_3$, 85 °C, 36 h (33%); (b) Pd(PPh$_3$)$_2$Cl$_2$, CuI, Trimethylsilylacetylene, Et$_3$N, 85 °C sealing-tube, 16 h (78%); (c) TBAF (soln 1M in THF), THF, 20 °C, 1 h; (d) Pd(PPh$_3$)$_2$Cl$_2$, CuI, DMF/NEt$_3$, 80 °C, 16 h (30%).

*Cyclopentadithiophene Derivatives.* The known cyclopentadithiophene **8** [32] was di-iodinated at the C2 and C6 positions using *N*-iodosuccinimide, affording **9** in a 89% yield. Double Sonogashira coupling between **9** and the terminal alkyne **10** [33] afforded the push–push derivative **2a** in 45% yield. The pull–pull analogue **2b** was obtained in a 50% yield from **9** and the alkynylated sulfone **11** [25] after the in situ removal of the TMS protecting group with TBAF, leading to **2b** in a 50% yield (Scheme 3). These derivatives are much more soluble in organic solvents than their bithiophene homologues.

### 2.2. Photophysical Properties

The photophysical properties of the new quadrupoles **1a,b** and **2a,b** and the related figures of merit are collected in Tables 1 and 2.

**Scheme 3.** Synthesis of **2a,b** compounds. Reagents and conditions: (a) *N*-iodosuccinimide, THF (89%); (b) Pd(PPh$_3$)$_2$Cl$_2$, CuI, NEt$_3$, 45 °C, 16 h (45%); (c) TBAF, Pd(PPh$_3$)$_2$Cl$_2$, CuI, NEt$_3$, 45 °C, 16 h (50%).

**Table 1.** Photophysical properties of quadrupoles **1a,b** and **2a,b** vs. those of **3a** and **3b′** in toluene (unless otherwise stated).

| Cmpd | $\lambda_{abs}$ (nm) | $\varepsilon_{max}$ (M$^{-1}$ cm$^{-1}$) | $\lambda_{em}$ (nm) | $\Delta\omega$ [a] (cm$^{-1}$) | $\phi_F$ [b] | $\phi_\Delta$ [c] | $1-\phi_F-\phi_\Delta$ | $\varepsilon_{max}\,\phi_F$ [d] (M$^{-1}$ cm$^{-1}$) | $\sigma_2^{max}$ [e] (GM) | $\sigma_2^{max}\,\phi_F$ [f] (GM) | $\sigma_2^{max}/M$ [g] (GM.mol/g) | $\sigma_2^{max}/(N_{eff})^2$ (GM) | $\sigma_2\,\phi_\Delta$ (GM) |
|---|---|---|---|---|---|---|---|---|---|---|---|---|---|
| **1a** | 414 305 | 65,000 33,900 | 475, 507 | 3102 | 0.35 | 0.54 | 0.11 | 22,750 | 900 | 315 | 1.831 | 1.236 | 486 |
| **1b** | 413 | 50,200 | 467, 496 | 2800 | 0.28 | 0.65 | 0.07 | 14,056 | 610 | 171 | 1.537 | 0.770 | 397 |
| **2a** | 438 303 | 69,800 30,500 | 485, 517 | 2212 | 0.34 | 0.49 | 0.17 | 23,732 | 1300 | 442 | 1.910 | 1.786 | 637 |
| **2b** | 446 | 59,200 | 496, 520 | 2260 | 0.39 | 0.39 | 0.22 | 23,088 | 1510 | 589 | 2.649 | 1.911 | 589 |
| **3a** [h] | 390 301 | 126,800 55,000 | 428 | 2277 | 0.85 | nd [i] | <0.15 | 107,780 | 980 | 833 | 1.574 | 1.047 | / |
| **3b′** [j] | 372 | 69,200 | 404 | 2129 | 0.98 | nd [i] | <0.02 | 67,816 | >68 [k] | >67 | >0.132 | >0.066 [k] | / |

[a] Stokes shift. [b] Fluorescence quantum yield in toluene. [c] Singlet oxygen formation quantum yield determined relatively to tetraphenylporphyrin in toluene ($\phi_\Delta$[TPP] = 0.70). [d] One-photon brightness. [e] Maximal intrinsic 2PA cross-sections in toluene at maximum, measured by TPEF in the femtosecond regime. A quadratic dependence of the fluorescence intensity on the excitation power is observed and the 2PA responses are fully non-resonant. [f] Two-photon action cross-section. [g] Molecular weight-scaled two-photon cross-section (*M* is the molecular mass expressed in g.mol$^{-1}$). [h] Data collected in CH$_2$Cl$_2$ [24]. [i] Not determined. [j] Nonyl groups on the fluorenyl instead of butyl groups [25]. [k] Only the forbidden 2PA band was detected (see text). The allowed band maximum is located at wavelengths lower than the detection limit (705 nm).

**Table 2.** Fluorescence lifetime ($\tau$), radiative ($k_R$) and non-radiative ($k_{NR}$) decay constants, and the theoretical fluorescence lifetime in the absence of other deactivation processes ($\tau_F^{DFT}$) for **1a,b**, **2a,b** and **3a,b′** in toluene (unless otherwise specified).

| Cmpd | $\tau$ (ns) | $k_R$ (s$^{-1}$) | $k_{NR}$ (s$^{-1}$) | $\tau_R^{DFT}$ [a] (ns) |
|---|---|---|---|---|
| **1a** | 0.29 | $1.2 \times 10^9$ | $2.2 \times 10^9$ | 1.54 |
| **1b** | 0.27 | $1.0 \times 10^9$ | $2.7 \times 10^9$ | 1.63 |
| **2a** | 0.56 | $0.6 \times 10^9$ | $1.2 \times 10^9$ | 1.64 |
| **2b** | 0.80 | $0.5 \times 10^9$ | $0.8 \times 10^9$ | 1.81 |
| **3a** [b] | 1.10 | $0.8 \times 10^9$ | $0.1 \times 10^9$ | 2.08 |
| **3b′** [c] | 0.79 | $1.2 \times 10^9$ | $0.03 \times 10^9$ | 1.83 |

[a] Value obtained for the most stable conformer (*a*) of each compound (Table 3). [b] Data collected in CH$_2$Cl$_2$ [24]. [c] For **3b′** (nonyl chains on the fluorenyl instead of butyl ones) [25].

**Table 3.** CAM-B3LYP calculations for the four lowest-energy excitations of the most stable form (*a*) in toluene (PCM). $E_{0n}$ is the transition energy (in eV), $\lambda_{calc}$ is the wavelength (in nm) and $f_{0n}$ is the oscillator strength of the absorption transition.

| Cmpd | $S_{0n}$ | $E_{0n}$ | $\lambda_{calc}$ | $f_{0n}$ | Main MO Transition Percentage |
|---|---|---|---|---|---|
| **1a** | 1 | 2.88 | 433 | 2.89 | HOMO → LUMO (79%) |
| | 2 * | 3.50 | 354 | 0.008 | HOMO-1 → LUMO (48%) |
| | 3 | 3.97 | 313 | 0.37 | HOMO-2 → LUMO (48%) |
| | 8 | 4.33 | 286 | 0.54 | HOMO-1 → LUMO + 6 (26%) |
| | | | | | HOMO-1 → LUMO + 5 (17%) |
| | 27 | 5.26 | 236 | 0.23 | HOMO-11 → LUMO (42%) |
| **1b** | 1 | 2.74 | 426 | 2.44 | HOMO → LUMO (89%) |
| | 2 * | 3.74 | 331 | 0.003 | HOMO → LUMO + 1 (65%) |
| | 4 | 4.48 | 277 | 0.38 | HOMO → LUMO + 2 (58%) |
| | | | | | HOMO-1 → LUMO + 1 (21%) |
| | 10 | 4.97 | 250 | 0.23 | HOMO-3 → LUMO (33%) |
| | 18 | 5.55 | 223 | 0.26 | HOMO → LUMO + 3 (53%) |
| **2a** | 1 | 2.73 | 454 | 2.80 | HOMO → LUMO (85%) |
| | 2 * | 3.51 | 353 | 0.13 | HOMO-1 → LUMO (46%) |
| | 3 | 3.95 | 314 | 0.43 | HOMO-2 → LUMO (40%) |
| | | | | | HOMO-1 → LUMO + 1 (28%) |
| | 8 | 4.31 | 288 | 0.55 | HOMO-1 → LUMO + 6 (42%) |
| | 20 | 5.02 | 248 | 0.24 | HOMO-5→ LUMO (58%) |
| **2b** | 1 | 2.86 | 455 | 2.40 | HOMO → LUMO (90%) |
| | 2 * | 3.64 | 340 | 0.11 | HOMO → LUMO + 1 (72%) |
| | 3 | 4.42 | 281 | 0.37 | HOMO → LUMO + 2 (73%) |
| | 6 | 4.77 | 260 | 0.22 | HOMO-2 → LUMO (77%) |
| | 18 | 5.42 | 229 | 0.22 | HOMO → LUMO + 4 (47%) |
| | | | | | HOMO-6 → LUMO + 1 (28%) |

\* These transitions correspond to the first forbidden 1PA transition, which is 2PA allowed.

*One-Photon Absorption.* The 1PA spectra of **1a,b** and **2a,b** are very similar (Figure 1), with a strong band at the edge of the UV-visible range (410–450 nm) and a second, less intense, band at higher energy. For the diphenylamino derivatives, this second absorption band is at ca. 300 nm, while for the triflate derivatives **1b** and **2b**, the maximum is located outside the observation window (below 290 nm), consistent with the TD-DFT calculations (see later). As discussed below, the lowest energy band corresponds to the HOMO-LUMO transition (Table 3). Regardless of the nature of the central core, this transition has some charge-transfer (CT) character, from the peripheral (electron-rich) branches towards the center for the amino derivatives **1,2a**, and from the core to the periphery for the triflate derivatives **1,2b**. For a given substituent set, replacing the bithiophene core by the cyclopentadithiophene one (i.e., going from **1** to **2**) always leads to a red shift of the first absorption band, regardless of the substituent electronic effect.

*Emission Properties.* All compounds are fluorescent (Figure 1), with lifetimes less than a nanosecond. For an invariant endgroup, the bithiophene derivatives (**1a,b**) are less fluorescent than their cyclopentadithiophene analogues (**2a,b**). Within each family of compounds (**1a,b** vs. **2a,b**), there is no clear trend in the electronic substituent influence on the quantum yield. The emission bands are vibronically structured with smaller Stokes shifts ($\Delta\omega$) for **2a,b** than for **1a,b**, consistent with a lower structural reorganization following excitation for the cyclopentadithiophene derivatives. We return briefly to the origin of the emission bands' vibrational fine structure in the ESI (Figure S15). The fluorescence lifetimes ($\tau$) were then measured for all these compounds, and the radiative ($k_R$) and nonradiative ($k_{NR}$) decay rates were derived (Table 2). The radiative decay rates for **1a,b**, **2a,b** and **3a,b** are similar, indicating that the differences in the fluorescence yields are mostly due to a non-radiative decay processes. In accordance with that finding, the DFT computations (see Section 2.3) also suggest that, regardless of the conformation considered, **1a,b** and **2a,b** should have fairly similar fluorescence lifetimes in the absence of any other (nonradiative) process.

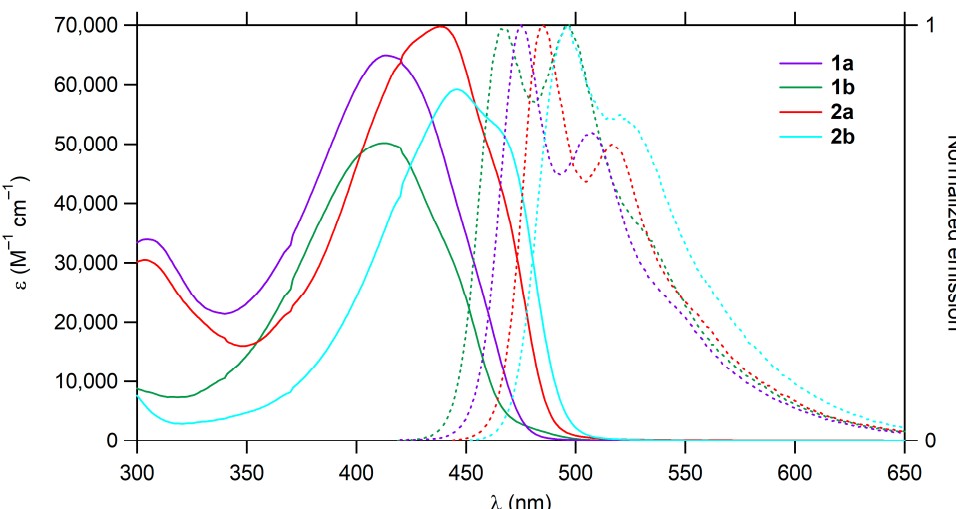

**Figure 1.** Absorption (solid lines) and emission (dashed lines) spectra of quadrupoles cores **1a,b** and **2a,b** in toluene.

*Singlet Oxygen Production.* Both the bithiophene (**1a,b**) and the cyclopentabithiophene (**2a,b**) derivatives photosensitize oxygen, the former compounds being more efficient at producing singlet oxygen under 1PA conditions, according to the measured quantum yields in toluene ($\phi_\Delta$: Table 1). Interestingly, based on the $\sigma_2$ $\phi_\Delta$ figures of merit, under 2PA conditions, the cyclopentabithiophenes (**2a,b**) are more efficient than the bithiophenes (**1a,b**). Given that the yields of all the photochemical processes subsequent to excitation should amount to unity, the yield of the excited singlet molecule potentially undergoing internal conversion (or any other non-radiative processes except oxygen photosensitization) can be inferred for **1a,b** and **2a,b** by computing the difference $1-\phi_F-\phi_\Delta$ (Table 1). All the percentages found are below 25%, indicating that internal conversion is minimal for all these compounds. The $1-\phi_F$ values derived for their fluorene analogues (**3a,b**) reveal that less than 15% of the fluorescent excited state undergoes non-radiative decay (i.e., intersystem crossing and internal conversion) in air (i.e., in the presence of oxygen), suggestive of an even lower quantum yield for potential singlet oxygen formation ($\phi_\Delta$).

*Two-Photon Absorption.* Two-photon excited florescence measurements (TPEF) reveal that **1a,b** and **2a,b** give rise to significant 2PA (Table 1 and Figure 2). When progressing from bithiophenes **1a,b** to cyclopentadithiophene homologues **2a,b**, and regardless of the nature of the peripheral substituent, an increase in $\sigma_2$ is obtained, consistent with a positive impact of strapping the two central thienyl rings together. This result can be traced back to the structure of the cyclopentadithiophene spacer and not to the number of effective electrons ($N_{eff}$), which remains constant for **1a** and **2a** (27.0) and for **1b** and **2b** (28.1) [34]. Of the cyclopentadithiophene derivatives **2a,b**, the better $\sigma_2$ value is obtained for **2b**, which features electron-withdrawing groups at the periphery, in combination with the increased electron-richness of the central core proceeding from **1a,b** to **2a,b**, and as supported by the DFT calculations (see Section 2.3) [28].

Superimposing 1PA spectra plotted at twice the wavelength on the 2PA spectra can aid in identifying the excited state that is populated by a given 2PA transition of interest. In the case of **1a,b** and **2a,b** (ESI, Figure S8), the first 2PA band appears at shorter wavelengths than those of the first allowed 1PA state for these compounds. Consistent with the three-level model for related quadrupolar systems [35] and with TD-DFT and SAOP calculations in the present study (see Section 2.3 below), this 2PA band likely originates from the population of a symmetric (*g*-type) excited state located at slightly higher energy than the antisymmetric (*u*-type) state corresponding to the strongly allowed, lowest-energy 1PA band. From the exclusion rule operative for centrosymmetric compounds, this second singlet excited state (S$_2$) populated by 2PA is, in principle, 1PA-forbidden, and should not be observed in the 1PA spectra (although a weak shoulder can still be observed in

1PA), explaining the apparent shift between the 1PA and 2PA peaks. Interestingly, this model also seems to apply to **2a,b** which are not centrosymmetric ($C_{2v}$ symmetry with ideal conformations), although they possess a significant quadrupolar character along their long axis, similar to **3a,b** [28,36]. The shoulder on the low energy side of the first 2PA absorption of **1a,b** and **2a,b** corresponds to population of the lowest (*u*-type) excited state, a transition in principle 2PA-forbidden under strict centrosymmetry, with the 2PA cross-sections of ca. 120–140 GM for **1a,b** and ca. 60–150 GM for **2a,b** indicating some relaxation of the exclusion rule for these compounds in a toluene solution.

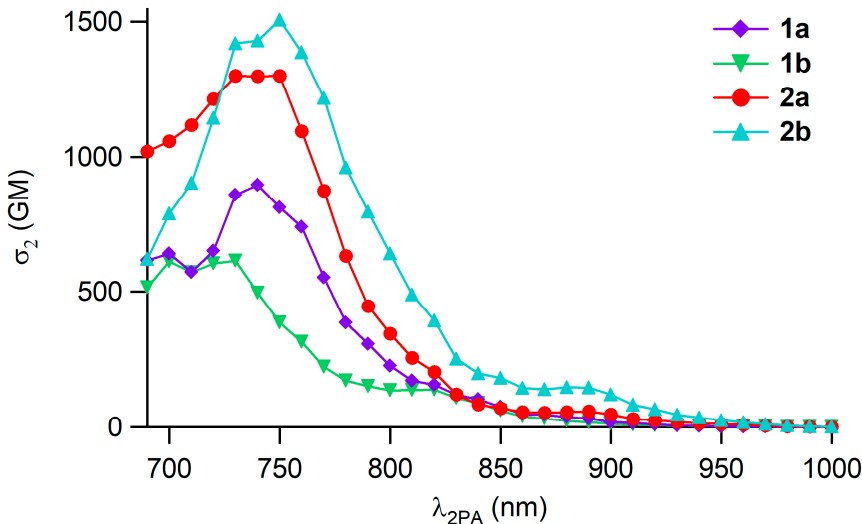

**Figure 2.** Two-photon absorption spectra of **1a,b** and **2a,b** in toluene.

Finally, the $\sigma_2$ value for **1a** is comparable to that of its corresponding fluorene-based homologue **3a** [37]; the absolute value is slightly less and the number-of-effective-electrons ($\sigma_2/(N_{eff})^2$) scaled value [38] is slightly more. The presence of 2,5-thienyl rings in place of 1,4-phenylene groups reinforces the electron-releasing power of the central unit, disfavoring a large charge transfer (CT) during the 2PA transition when electron-releasing arms are appended to this core, a feature that may also disfavor multiphotonic absorption. The $\sigma_2$ value of **2a** is superior to that of **3a**, indicating that planarization is beneficial for 2PA in these compounds. Thus, although more effective electrons ($N_{eff}$) are present in **3a** (30.6) than in **1a**, **2a** (27.0), larger 2PA cross-sections are obtained for **2a** [34]. Furthermore, **2a,b** display the best molecular weight-scaled 2PA cross-sections ($\sigma_2^{max}/M$) of the compounds in the present study (Table 1), highlighting the potential of this particular core for the design of efficient two-photon absorbers [39].

### 2.3. DFT Studies

DFT calculations were conducted in order to learn more about the electronic structures of **1a,b** and **2a,b** (Scheme 1) and their relation to their optical properties of interest. We will first report the ground state (GS) characteristics, i.e., the geometric and electronic structures of the aforementioned compounds after geometry optimizations. Next, their one-photon absorption (1PA) and emission properties from the TD-DFT calculations will be summarized. Finally, the results from simulating the 2PA spectra will be analyzed (see experimental for details of our approach). The related theoretical data are already available for **3a,b** [28]. The purpose of this systematic approach is to theoretically derive a complete set of structure–property relationships for the compounds shown in Scheme 4 and to reproduce and understand the experimental trends in order to identify the key electronic factors that favor 2PA, luminescence and singlet oxygen generation.

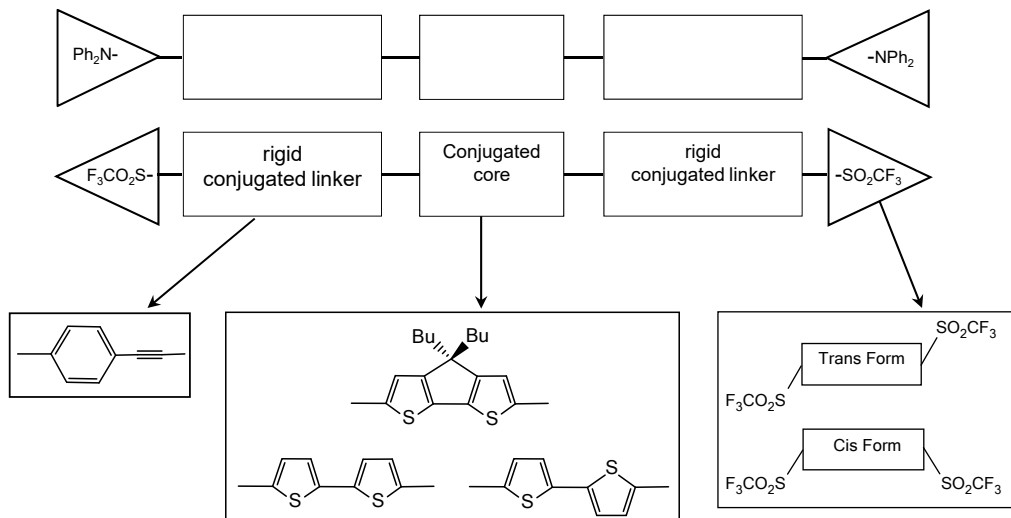

**Scheme 4.** Overview of molecular structures **1a,b, 2a,b** and **3a,b** investigated for their photophysical properties (**a** and **b** labels for the NPh$_2$ and SO$_2$CF$_3$ terminal groups, respectively).

*Ground State Geometries.* Different possible geometric configurations and conformations of **1a,b** and **2a,b** were explored, with minima found for all the compounds after optimization at the PBE0/6-31 + G(d) level in toluene. We anticipated that the bithiophenes **1a,b** might coexist as *cis* and *trans* conformations, which was confirmed, but other conformers with varying orientations for the triflate groups were also identified for **1b** and **2b**, as well as two forms with differing orientations of the butyl groups with respect to the π-conjugated backbone for **2a**. In total, eleven distinct GS conformations were identified: two for **1a** and **2a**, four for **1b**, and three for **2b** (see ESI, Table S1 and Figure S1 for their relative energies and structural arrangements, respectively). The geometry of the most stable conformer (*a*) for each compound is shown in Figure 3. All belong to the C$_s$ symmetry group and none of them presents a higher symmetry. For a given compound, the computed bonding parameters do not change significantly between the conformers and remain very close to the experimental expectations (see ESI, Tables S1 and S2) [40]. Given their energy differences, all these conformers (apart from, perhaps, the isomer *c* of **2b**) should coexist in a solution, including, in particular, the *cis* and *trans* isomers of **1a** and **1b**. The photophysical properties (Figure 4) found for these different GS conformers were very similar with no significant differences in the electronic transitions (see ESI, Tables S3–S5). Therefore, unless specified, the computational results discussed hereafter will concern only the most stable of them (isomers *a* in Figure 3), which should also correspond to the dominant conformer present in a solution at ambient temperatures (Figures 5 and 6).

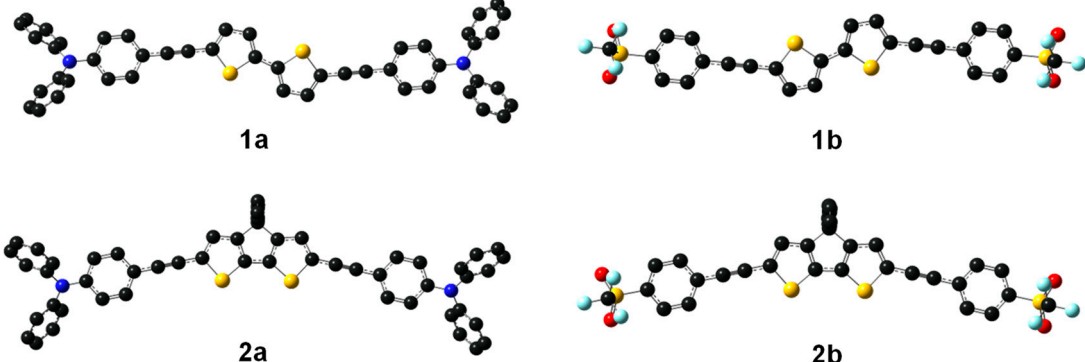

**Figure 3.** DFT PBE0/6-31 + G(d) optimized structures of the most stable GS conformers for molecules **1a,b** and **2a,b** in toluene.

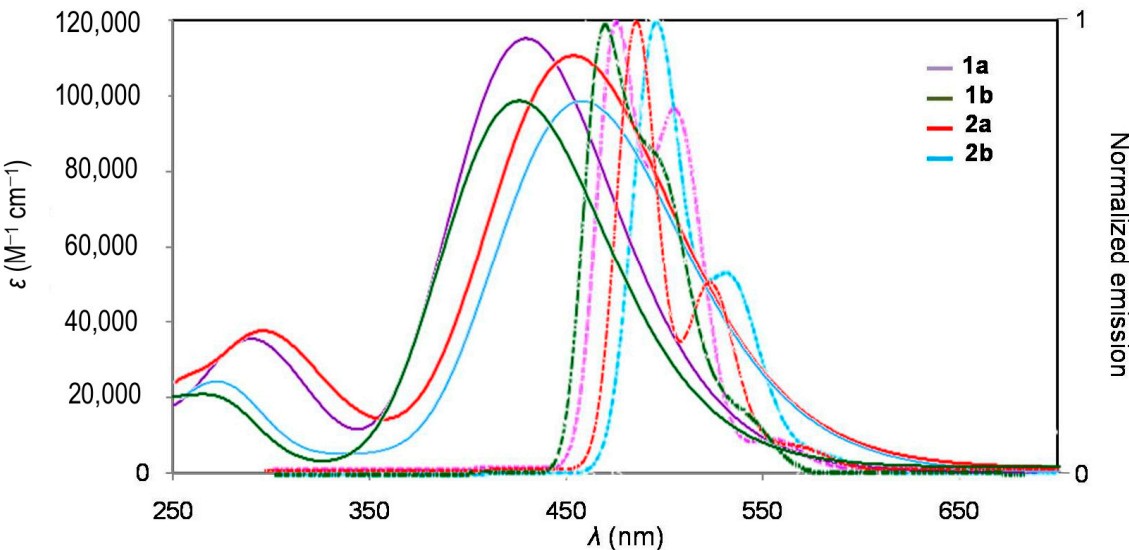

**Figure 4.** CAM-B3LYP/6-31 + G(d) simulated UV-vis absorption (solid line) and emission (dashed line) spectra (bottom) of **1a**,**b** and **2a**,**b** in toluene.

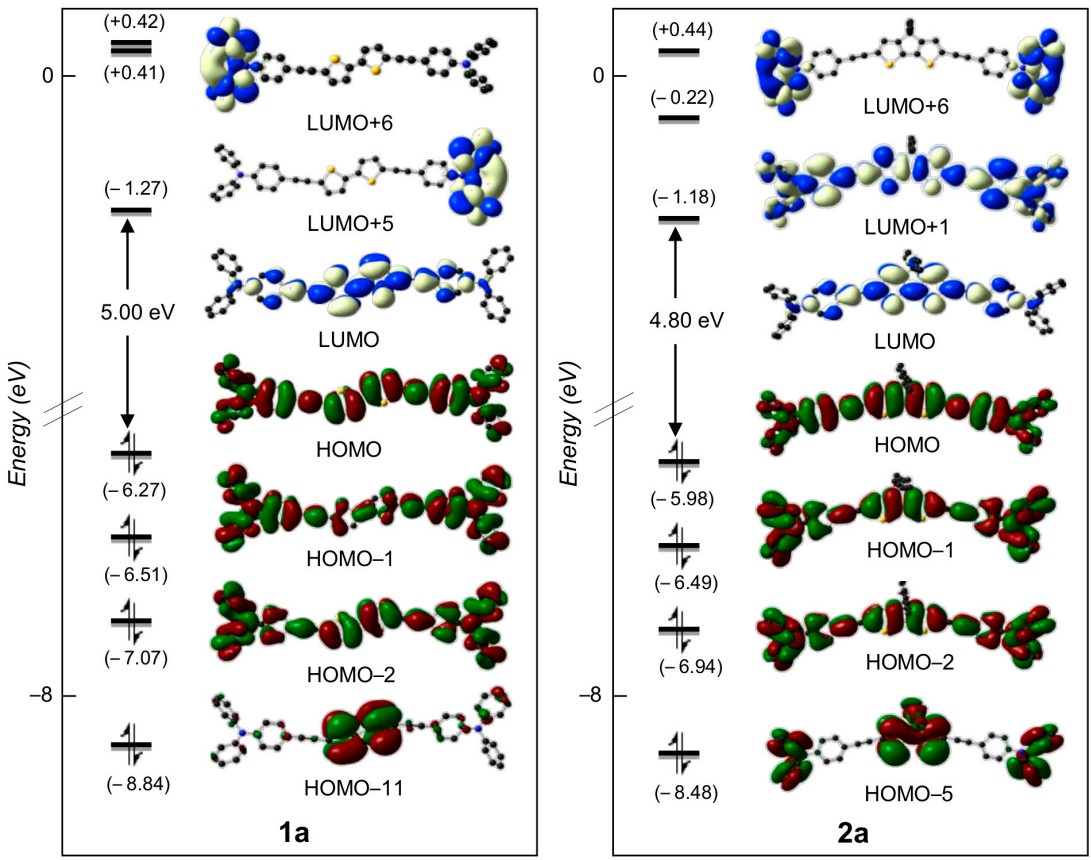

**Figure 5.** Frontier MO diagrams of **1a** (**left**) and **2a** (**right**) (contour isodensity values: $\pm 0.015$ (e/bohr$^3$)$^{1/2}$). The hydrogen atoms are omitted for clarity.

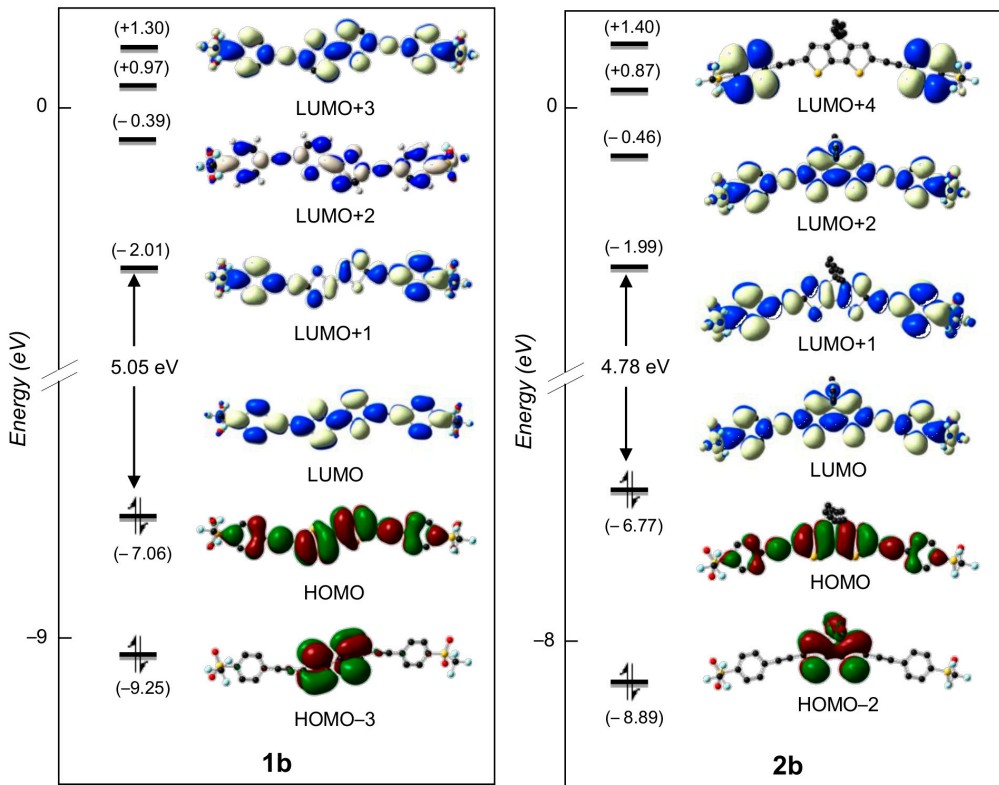

**Figure 6.** Frontier MO diagrams of **1b** (**left**) and **2b** (**right**) (contour isodensity values: ±0.015 (e/bohr$^3$)$^{1/2}$). The hydrogen atoms are omitted for clarity.

*Structural Effects and Electronic Structure.* Overall, the electronic structures of **1a**,**b** and **2a**,**b** are very similar. For **1a** and **2a** with electron-releasing substituents, the HOMOs (π orbitals) extend over the whole compound (including the peripheral branches) except for the sulfur atoms, whereas the LUMOs are π*-type MOs that are more localized on the central aromatic rings. For the triflate analogues **1b** and **2b**, the reverse trend is observed, with the LUMOs now extending over all of the molecule. Again, the sulfur atoms are not involved in the HOMOs. The main effect of planarization when progressing from **1a**,**b** to **2a**,**b** is to reduce the HOMO-LUMO gap. Despite the sulfur atoms not being included in the HOMO, they do have an impact compared to the fluorenyl reference compounds **3a**,**b**; the sulfur atoms electronically enrich the central aromatic rings, resulting in a clear decrease of the first oxidation potential. While computations aimed at determining the oxidation potentials for **1b**, **2b** and **3b** (lacking an oxidizable substituent) did not converge, the literature data can provide guidance. Based on the data for 2,2′-bithiophene **12** (1.00–1.20 V vs. SCE) [41], for the cyclopentadithiophenes **13a**,**b** (ca. 1.00 V vs. SCE) [42] and the fluorene **14** (ca. 1.70 V vs. SCE) [43] in CH$_3$CN (Scheme 5) [44], the replacement of the 1,4-phenylene rings of **3a**,**b** by 2,5-thienyl groups is expected to result in a decrease in the first oxidation potential of the central aromatic unit in **2a**,**b** (of ca. 1 V), while **1a**,**b** should have an oxidation potential located in between these values [45].

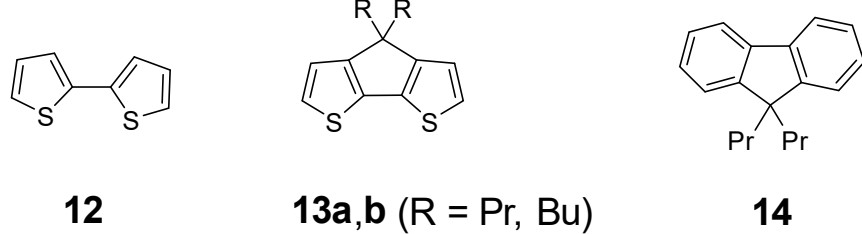

**Scheme 5.** Selected monomers previously subjected to electropolymerization studies in CH$_3$CN.

*One-Photon Absorption Properties.* The photo-physical properties of **1a,b** and **2a,b** (Scheme 1) were then studied. The TD-DFT computations were carried out on the most stable conformers (*a*) of all the compounds with different electron-releasing (X = NPh$_2$) or -accepting (X = SO$_2$CF$_3$) groups in order to assign the different absorption bands that were experimentally observed in toluene (Table 1 and Figure 1). Starting from the PBE0/6-31 + G(d)-optimized structures, the CAM-B3LYP calculations lead to a near-perfect agreement with the experiment (Table 3), whereas PBE0 systematically overestimated the experimentally measured wavelengths (ESI, Table S3). Hereafter, only the CAM-B3LYP results are discussed. The simulated absorption spectra are shown in Figure 4, and the molecular orbitals (MOs) involved in the corresponding electronic excitations are displayed in Figures 5 and 6.

In the case of **1a,b** and **2a,b** featuring peripheral electron-donating (a) or electron-withdrawing (b) groups, the lowest energy absorption corresponds to an electronic transition from the HOMO to the LUMO. The charge transfer that occurs during excitation is from the electron-rich to the electron-poorer parts and is therefore related to the molecular structures, namely from the periphery to the center of the molecule in **1a** and **2a** and the reverse in **1b** and **2b**, similar to what was previously computed for **3a,b** [28]. The two donor groups (NPh$_2$) in **1a** and **2a** destabilize the HOMO, which becomes more substituent-centered than in **1b** and **2b**, while the two accepting groups (SO$_2$CF$_3$) in **1b** and **2b** stabilize the LUMO, which become more substituent-centered than in **1a** and **2a**. For these cases, a similar HOMO-LUMO energy gap reduction results compared to that of the unsubstituted diphenylethynyl analogue. This effect is stronger in the cyclopentadithiophenes **2a,b**, possibly because of the forced coplanarity of the thiophenes that favors π-interactions with the peripheral substituents. Overall, the HOMOs and LUMOs are of the π/π*-type character and are responsible for the lowest energy absorption bands, which are delocalized over the molecular backbones with important contributions at the thiophenes (excluding the sulfur atoms for the HOMOs). At a higher energy, the second absorption band (in the range 277–314 nm) results from the overlap of several weakly allowed transitions, the lowest energy of which is a HOMO-2 → LUMO transition for compounds with peripheral electron-releasing substituents (**1a** or **2a**) and a HOMO → LUMO + 2 transition for compounds with peripheral electron-accepting substituents (**1b** or **2b**). To ascertain the nature of the charge-transfer taking place into the S$_1$ excited state, we have computed the difference between the electron densities of the vertical S$_1$ and S$_0$ states of these molecules in toluene (Figure 7). The change in the total electron density for the ground and vertical excited states [$\Delta\rho(r) = \rho_{S1}(r) - \rho_{S0}(r)$] [46] depicts the direction of the charge transfer upon excitation from S$_0$ to S$_1$ (Figure 7). A very short charge transfer distance $d^{CT}$ between the positive and negative charge centroids is observed due to the high symmetry of the molecules. Nevertheless, the amount of charge $Q^{CT}$ transferred between S$_0$ and S$_1$ is important (ca. 0.5 e) in the four molecules and depends on the nature of the peripheral substituents. As expected, it is slightly larger in the cyclopentadithiophenes derivatives, most likely because of the planarity of these molecules favoring a better π-overlap.

*One-Photon Emission Properties.* The fluorescence wavelengths were deduced from the energy difference between the S$_1$ relaxed excited state and the ground state S$_0$. Again, a fair match with the experiment is observed (Figures 1 and 4) with the fluorescence spectra for **1a,b** and **2a,b** exhibiting the same pattern, i.e., an intense band with a lower-energy intense shoulder, nicely reproducing the measured spectrum. To fully assess the validity of our computational approach for studying the excited states, fluorescence lifetimes ($k_F$), Stokes shifts and quantum yields were also calculated (ESI, Table S6). For these data, the match with the experiment is only fair, but the trends in behavior for the different families of fluorophores featuring electron-donor or -acceptor peripheral substituents are reproduced. Thus, the Stokes shift for the bithiophenes (**1a,b**) are 2246 $\pm$ 200 cm$^{-1}$ vs. 1620 $\pm$ 70 cm$^{-1}$ for their cyclopentabithiophene homologues (**2a,b**). This suggests a smaller structural and solvation-related change upon excitation for the latter, readily explicable due to the presence of the rigidifying 1,1′-dibutylmethylene bridge. Interestingly, the geometries

of the $S_0$ and $S_1$ excited states of these compounds were computed to be rather similar, leading to good overlap between the normal modes and allowing for a vibronic treatment to be performed (see ESI, Figure S15). Analysis of these vibrational modes indicates that the lowest excited states of **1a**,**b** and **2a**,**b** show structural changes mostly located on the thiophene rings, in line with its assignment as a $\pi \rightarrow \pi^*$ transition in these derivatives.

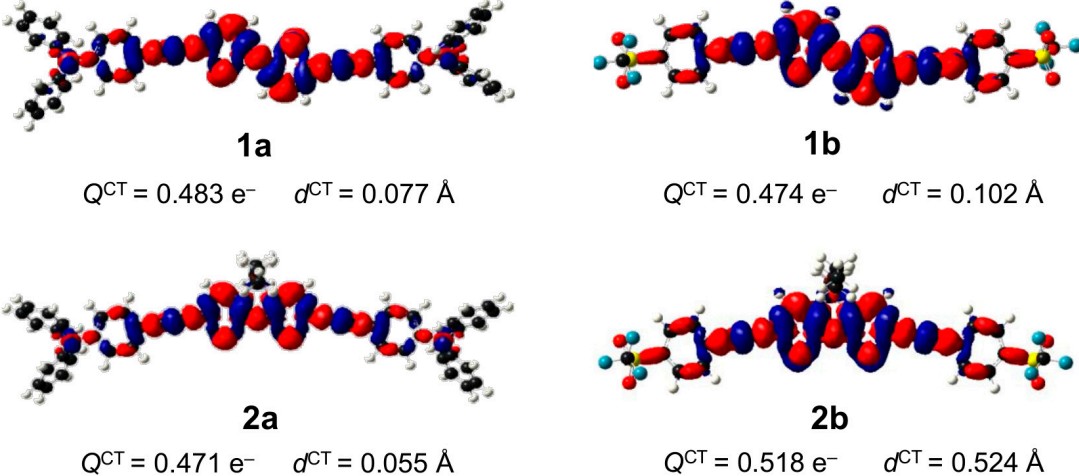

**1a**

$Q^{CT} = 0.483$ e⁻     $d^{CT} = 0.077$ Å

**1b**

$Q^{CT} = 0.474$ e⁻     $d^{CT} = 0.102$ Å

**2a**

$Q^{CT} = 0.471$ e⁻     $d^{CT} = 0.055$ Å

**2b**

$Q^{CT} = 0.518$ e⁻     $d^{CT} = 0.524$ Å

**Figure 7.** Change in total density (computed at the CAM-B3LYP/6-31 + G(d) level in toluene) upon excitation from the ground state $S_0$ to the vertical $S_1$ excited state for **1a**,**b** and **2a**,**b**. The red and blue colors indicate an increase and a decrease of electron density, respectively (contour isodensity value: 0.0002 e/bohr³). $Q^{CT}$ and $d^{CT}$ are the amount of transferred charge and the distance between the centroids of the positive and negative charges, respectively.

*Triplet Excited States.* Finally, given the importance of the triplet states for oxygen sensitization [47], we have computed the energies of the lowest lying triplet states $T_1$ and $T_2$ for the most stable conformers of these molecules, the results revealing that the $T_1$ states will lie between 0.49 (**2a**) and 0.86 eV (**2b**) below the corresponding $S_1$ states (Figure 8). Consequently, intersystem crossing will most likely occur via $T_2$, which is energetically located between $T_1$ and $S_1$. From their $T_2$ states, these compounds will then internally convert to $T_1$ before decaying to the GS or reacting with oxygen. The decay from $T_1$ to the GS must be largely non-radiative because no phosphorescence was detected for **1a**,**b** and **2a**,**b** at ambient temperatures. It is indeed favored by the low-energy gap rule [48], since all the $S_0$-$T_1$ energy gaps are lower than 1.8 eV. Interestingly, in spite of the reduced HOMO-LUMO gaps of **2a**,**b** compared to **1a**,**b** (see above), the energies of the $T_1$ states of **2a**,**b** are not lower than those of **1a**,**b**, but actually bracket their energies.

In order to map the main localization of unpaired electrons in these $T_1$ triplet states, their spin distribution was also computed for **1a**,**b** and **2a**,**b** (ESI, Figure S16). For all these molecules, the positive spin density accumulates around both central C=C-S units, adopting a Z-shaped distribution in the *trans*-bithiophenes **1a**,**b** and a U-shaped distribution in the cyclopentadithiophenes **2a**,**b**, while the negative spin densities alternate with positive spin densities on the $\pi$-manifolds of the peripheral phenylethynyl arms. There is no localization of the spin density on the butyl groups, triflate groups or peripheral phenyls of the diphenylamino substituents. For both families of compounds, the electron-releasing (diphenylamino) substituents result in increased spin density alternating along the $\pi$-manifold in comparison to the electron-withdrawing groups. The calculations indicate that within each family (**1a**,**b** or **2a**,**b**), proceeding from the triflate to the diphenylamino-substituted derivative slightly increases the spin density in the central $\pi$-manifold of the bithiophene unit. For a given terminal substituent, replacing a bithiophene core with a cyclopentadithiophene core has the same effect. Thus, the maximal spin density in the $T_1$ state is found for **2a** and decreases in the order: **2a** > **1a** > **2b** ≈ **1b**, while the spin

density on the sulfur atom is slightly lower in **2a,b** than in **1a,b** and is independent of the peripheral substituent.

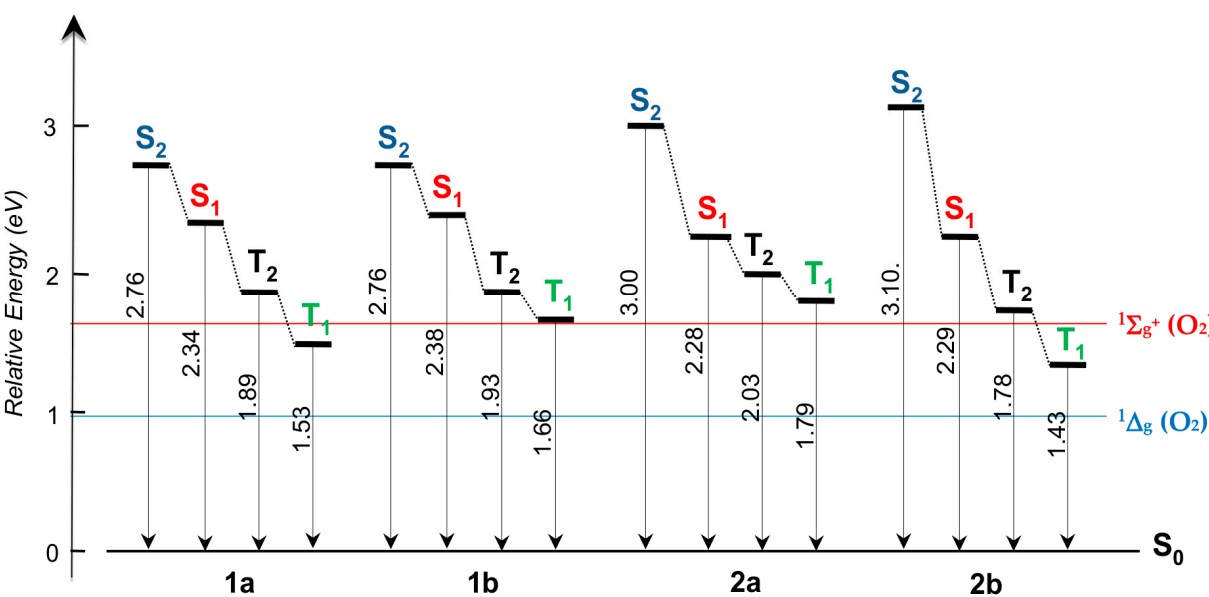

**Figure 8.** Relative energies (eV) of the excited states $S_1$, $S_2$, $T_2$ and $T_2$ with respect to that of $S_0$ for **1a,b** and **2a,b** (the most stable conformer *a*) computed at the CAM-B3LYP/6-31 + G(d) level in toluene. The energy of the two lowest-energy singlet states of molecular oxygen is indicated by the red and blue lines.

In the presence of oxygen, according to Kasha's rule [49], it is the $T_1$ state that will most likely sensitize oxygen by energy transfer. The energies of the $T_1$ states of these compounds are very close to that of the second singlet state of oxygen ($^1\Sigma_g^+$) at 1.62 eV (Figure 8), so that oxygen may be primarily photosensitized *via* this state, except (perhaps) for **1a** and (especially) **2b**, compounds for which the energy of $T_1$ is (much) too low and can only be transferred to the (lower-lying) singlet state of oxygen ($^1\Delta_g$) at 0.97 eV (Figure 8) [47]. The large gap to that state possibly results in a reduced intersystem crossing rate, which may explain the lower photosensitizing efficiency of **2b**. In this connection, it is noteworthy that the photosensitization ordering found for the other compounds (**1b > 1a > 2a**) roughly correlates with their energy gaps to the $^1\Sigma_g^+$ (O$_2$) level. In contrast, the changes in the spin densities computed for the $T_1$ states of **1a,b** and **2a,b** (ESI, Figure S16) appear not to be correlated with the oxygen photosensitizing yields of these compounds ($\Phi_\Delta$: Table 1). Finally, because the oxidation potentials of all of these compounds are significantly lower than 1.9 eV, the formation of a charge-transfer complex with O$_2$ might happen in a solution [47]. In such a case, the overall oxygen photosensitization mechanism will be much more complex, and possibly invalidate the previous analysis based on a simple through-space (Förster) energy-transfer process between **1a,b** or **2a,b** in their $T_1$ state and spatially distant oxygen [49].

*Two-Photon Absorption Properties.* The 2PA merit of **1a,b** and **2a,b** systems in their most stable conformations were computed at the SAOP/DZP level in a vacuum (see ESI, Figure S17 for their corresponding SAOP/DZP frontier MO diagrams). Their 1PA spectra have also been recalculated at this level of theory for comparison. The simulated 2PA profiles are displayed in Figure 9 where the computed cross-sections $\sigma_2$ are plotted as a function of the two-photon absorption wavelength. The experimental and theoretical 1PA and 2PA maximum energies and cross-sections are summarized in Table 4, while the complete computed data ($\sigma_2$ (GM), 1PA and 2PA energies $\omega$ (eV)) are given in Tables S7–S14 (ESI). As is often expected from the exclusion rule, the computed 2PA wavelength corresponds to twice the 1PA wavelength of the forbidden $S_2$ excited state, in line with the experimental

observations (see TPEF measurements in Section 2.2). For such compounds, the $S_2$ state is one-photon forbidden (or approximately forbidden in the case of small distortions to the ideal $C_s$ symmetry) but two-photon allowed, whereas the $S_1$ state is one-photon allowed and two-photon forbidden. The difference between one- and two-photon absorption properties for first two singlet excited states ($S_1$ and $S_2$) in quadrupolar chromophores is usually at the origin of the mismatch observed between the main 2PA absorption and the main 1PA absorption band, when the 1PA spectrum and the 2PA spectrum plotted at twice its wavelength are superimposed (see ESI, Figure S9). The forbidden transition to the $S_2$ state can also be retrieved from the previously calculated CAM-B3LYP excitations (Table 3; see footnote), but due to their very low oscillator strengths, the peaks corresponding to this transition are hidden between the much more intense peaks of the (allowed) $S_0 \rightarrow S_1$ transition in the simulated 1PA spectra (Figure 4).

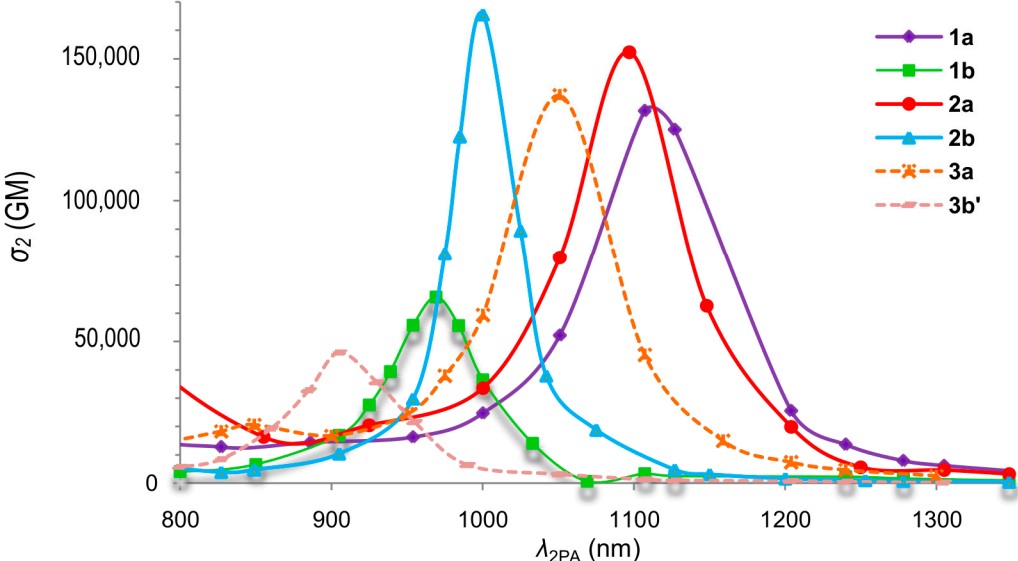

**Figure 9.** SAOP/DZP computed $\sigma_2$ vs. wavelength ($\lambda^{2PA}$) for the four molecules **1a,b**, **2a,b** and **3a,b** in toluene.

**Table 4.** Experimental and computed (for the most stable conformer *a*) OPA and TPA wavelengths $\lambda$ (nm), energies $\omega$ (eV) and cross-sections $\sigma_2$ (GM) of the molecules **1a,b**, **2a,b** and **3a,b**. The OPA oscillator strengths are between parentheses.

| Cmpd | Exp | | | | Theo [a] | | |
|---|---|---|---|---|---|---|---|
| | $\lambda_{max}^{TPA}$ | $\sigma_2^{max\,b}$ | $\omega^{OPA}$ | $\omega^{TPA}$ | $\lambda_{max}^{OPA}$ ($\varepsilon_{max}$) | $\lambda_{max}^{TPA}$ | $\sigma_2^{max\,b}$ |
| **1a** | 729 | 900 | 2.22 | 1.12 | 559 ($0.35 \times 10^{-3}$) | 1107 | 13,168 |
| **1b** | 729 | 610 | 2.55 | 1.28 | 486 ($0.17 \times 10^{-3}$) | 969 | 6560 |
| **2a** | 738 | 1300 | 2.25 | 1.13 | 551 ($0.41 \times 10^{-2}$) | 1097 | 15,241 |
| **2b** | 738 | 1510 | 2.49 | 1.24 | 498 ($0.12 \times 10^{-1}$) | 1000 | 16,579 |
| **3a** [c,d] | 720 | 980 | 2.35 | 1.18 | 528 ($0.42 \times 10^{-2}$) | 1051 | 13,710 |
| **3b′** [e]/**3b** [d] | 730 | / | 2.73 | 1.37 | 454 ($0.26 \times 10^{-2}$) | 905 | 7615 |

[a] Highest 2PA cross-section calculated in the 700–1350 nm range. [b] 1 GM = $10^{-50}$ cm$^4$ s photon$^{-1}$. [c] See: [24]. [d] See: [28]. [e] See: [25].

The computed $\sigma_2$ cross-sections for the bithiophenes (**1a,b**) and cyclopentadithiophenes (**2a,b**) are approximatively ten times higher than the measured ones. Such a large deviation between the experimental and SAOP/DZP-computed 2PA cross-sections has precedence [23,50], and has mainly been attributed to the underestimation of the energy gaps with the SAOP/DZP calculations [23]. Consistent with this, the computed 2PA wavelengths are longer than the experimental ones (ESI, Figure S17). The neglect of solvent

effects in the calculations is certainly responsible for part of this deviation [51]. Albeit overestimated, the computed 2PA cross-sections reproduce the experimental trends, i.e., the $\sigma^{2PA}$ values of **1a,b** and **2a,b** are ordered correctly, and thus, the theoretical computations confirm that, among this series, the cyclopentadithiophenes should exhibit the highest 2PA cross-sections.

Neglecting any differences in the reorganization energies of the first excited states, the energy difference between the $S_1$ and $S_2$ states should correspond to twice the effective excitonic/electronic coupling ($V$) between the branches through the central core (Figure 10) [52,53]. Such a coupling often induces a cooperative enhancement of the 2PA cross-section in these quadrupolar systems, which can be expressed according to Equation (1), obtained from the essential-state modelling of 2PA in quadrupolar systems (Figure 10) [23,35], where $\mu_{S1}$ and $\mu_{S2}$ are the transition moments from the GS toward the $S_1$ state and the $S_2$ state, respectively, $\Gamma_{S1}$ is the line broadening factor for the former transition and $E_{S1}$ and $E_{S2}$ are the respective energies of these states.

$$\sigma_2^{max} \propto (\mu_{S1})^2(\mu_{S2})^2/[\Gamma_{S1}(2E_{S1} - E_{S2})^2] = (\mu_{S2})^2(\mu_{S1})^2/[\Gamma_{S1}(E_{S1} - 2V)^2] \qquad (1)$$

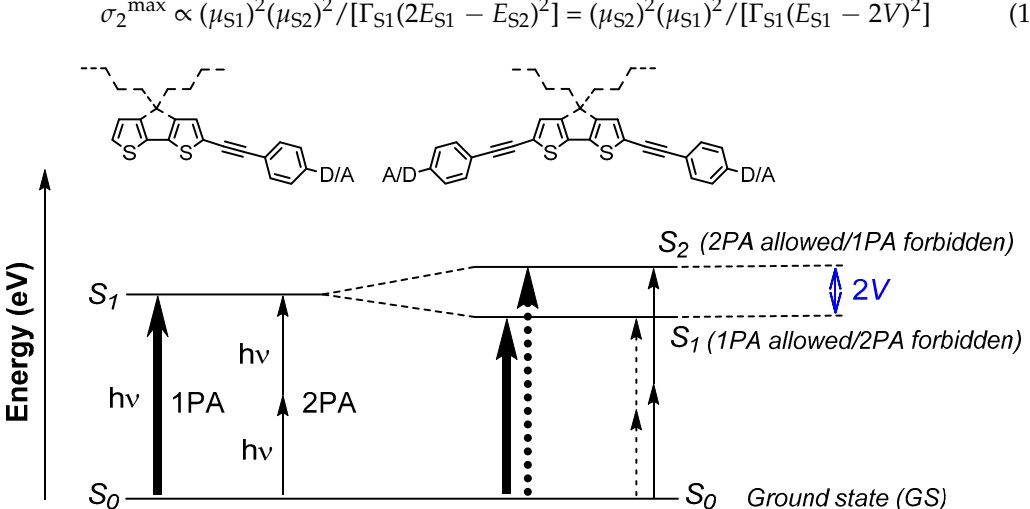

**Figure 10.** Change in allowed (plain arrows)/forbidden (dashed arrows) 1PA and 2PA transitions in the essential state models upon increasing branching in triphenylamine-based derivatives (D/A = electron-releasing/withdrawing groups).

Depending on the compound considered, this coupling amounts to 0.6–1.2 eV at the CAM-B3LYP level of theory. It is roughly double that of the experimental values, but qualitatively reflects the experimental trends previously derived (Table 5), except perhaps for **2b**, showing the limitations of the theoretical modelling.

**Table 5.** Experimental ($V_{exp}$) and CAM-B3LYP-computed ($V_{CAM-B3LYP}$) estimates from the effective excitonic/electronic ($V$) coupling from the available data for **1a,b**, **2a,b** and **3a,b/b′**.

| Cmpd | $\lambda_{1PA}$ | $\lambda_{2PA}{}^{max}$ | $\lambda_{2PA}{}^{min}$ [a] | $\sigma_2{}^{min}$ [a] | $V_{exp}$ | $V_{CAM-B3LYP}$ | | Ref. |
|---|---|---|---|---|---|---|---|---|
| | (nm) | (nm) | (nm) | (GM) | (cm$^{-1}$) | (cm$^{-1}$) | (eV) | |
| **1a** | 414 | 740 | 830 | 120 | 1436 | 2500 | 0.31 | This work |
| **1b** | 413 | 730 | 820 | 140 | 1592 | 4033 | 0.50 | This work |
| **2a** | 438 | 740 | 880 | 60 | 2098 | 4920 | 0.61 | This work |
| **2b** | 446 | 750 | 890 | 150 | 2123 | 3670 | 0.46 | This work |
| **3a** | 391 | 720 | 790 | 260 | 1101 | 1653 | 0.22 | [24] |
| **3b′/3b** | 372 | NA [b] | 730 [c] | 70 [c] | NA [b] | 2540 | 0.32 | [25] |

[a] Forbidden 2PA transition at the lowest energy (shoulder). [b] Not available, the maximum of the allowed 2PA band of **3b′** possibly lying outside the detector range (<705 nm). [c] Deduced from Figure 4 of ref. [25].

## 3. Discussion

The present work reveals that new derivatives **1a,b** and **2a,b** are interesting and potentially useful photophysical properties for bio-oriented applications, as they are emissive compounds that photosensitize oxygen.

*Conformers and Symmetry.* The DFT calculations suggest that different conformers with a coplanar central aromatic core will be present in a solution at ambient temperature. In terms of the linear optical properties (1PA, emission), the calculations also reveal that these mixtures should behave as if only the most stable conformer (*a*) was present. However, there is an important symmetry-related issue, because *syn* rotamers are likely to exist in a solution for the bithiophenes **1a,b**, and these non-centrosymmetric species may relax the exclusion rule regarding 2PA absorptions. Both the experimental and computational 1PA/2PA data collected for **1a,b** in the present study indicate that the exclusion rule is largely observed, similar to what was found for the non-centrosymmetric cyclopentabithiophene homologues **2a,b** ($C_{2v}$ symmetry with ideal conformations), for which no such rotamer concerns exist. Our results confirm that the exclusion rule will be obeyed for **1a,b** and **2a,b** regardless of the conformers considered, provided that the quadrupolar characteristic is maintained because this property essentially originates from the antisymmetry along the long axis of the molecule.

*One- and Two-Photon Absorption Properties.* **1a,b** and **2a,b** exhibit good light harvesting efficiencies due to their strong 1PA bands in the visible range. Co-planarization of the central bithiophene unit (proceeding from **1a,b** to **2a,b**) results in increased electronic delocalization between the central unit and the terminal substituents (Scheme 6) [5,42], increasing the transition moment (intensity) of the CT band at low energy when proceeding from **1a,b** to **2a,b**. Another effect of this co-planarization is a reduced HOMO-LUMO gap for the central unit in **2a,b** compared to **1a,b**, which induces a slight red-shift of the first allowed 1PA and 2PA bands, as well as a red-shift of the first emission band (Tables 1 and 4) for derivatives with different central cores but similar peripheral substituents.

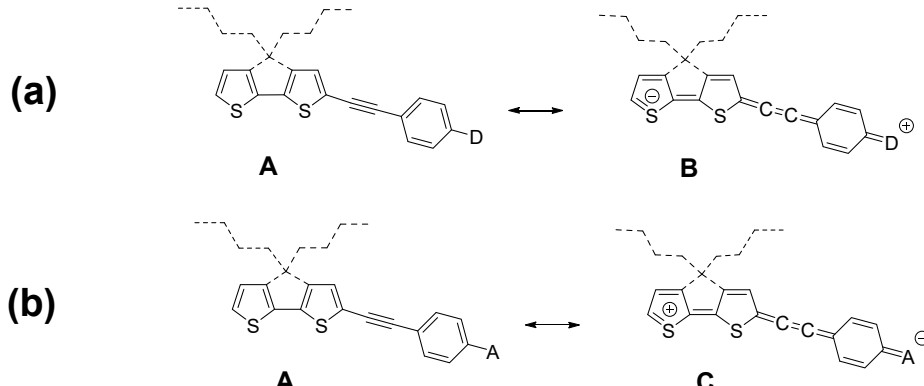

**Scheme 6.** Valence-bond mesomeric forms of in each dipolar part of the **1a,2a** (**a**) and **1b,2b** quadrupoles (**b**). The weight of the polar forms B and C is respectively larger in **2a,b** than in **1a,b** due to planarization of the π-manifold.

The 2PA cross-sections of **1a,b** are similar in magnitude to those previously reported for bithiophene derivatives **15** [18], **16** [15] and **17** (Scheme 7) [19]. The maximal 2PA cross-section presently found for **1a** (900 GM) is significantly larger than that of **15** (563 GM) [18,54], indicating that it is not only the electron-releasing power of the $NR_2$ substituents, as estimated by Hammett coefficients, that determines the 2PA response in these bithiophene derivatives; based on these substituent parameters, compound **15** with the dimethylamino ($NMe_2$) group would be expected to be more 2PA-efficient, as $NMe_2$ is more electron-releasing than $NPh_2$ [55,56].

**15**

$\sigma_2$ (700 nm) = 563 GM [DMSO]

**16**

$\sigma_2^{max}$ (760 nm) = 780 GM [toluene]

**18**

$\sigma_2^{max}$ (800 nm) = 693 GM [MeCN]

**17a**

$\sigma_2^{max}$ (795 nm) = 670 GM [DMSO]

**17b**

$\sigma_2$ (730 nm) = 210 GM [DMSO]

**17c**

$\sigma_2$ (796 nm) = 270 GM [TCE]

**Scheme 7.** 2PA cross-sections (in GM) measured by TPEF in a solution [solvent] at various wavelengths for some known bithiophenes.

The planarization of the central bithiophene core is beneficial for 2PA, the cross-sections ($\sigma_2^{max}$) increasing for homologous compounds featuring similar terminal substituents upon strapping the bithiophene unit (Table 1). The $\sigma_2^{max}/(N_{eff})^2$ figure of merit clearly reveals that this increase is not attributable to an increase in the number of active electrons (which are the same for **1a** and **2a** or between **1a** and **2b**), but rather that it results from the different central cores [38]. Based on the essential states models [35], this increase in 2PA can be understood as resulting from a larger photoinduced CT taking place in each of the dipolar branches that constitute these pseudo-quadrupolar derivatives (Scheme 8), and also an increase in the effective electronic/excitonic coupling (*V*) proceeding from **1a**,**b** to **2a**,**b** (Table 5). Planarization of the central unit favors a through-space interaction between the dipole moments of the CT in each half of the molecules (Scheme 8a) [57]. According to Equation (1) (see above), as long as $V \ll E_{S1}/2$, this increased effective coupling (*V*) will augment the cooperative behavior between the molecular halves and, as a result, the cross-section for the quadrupole will increase when going from **1a**,**b** to **2a**,**b**.

In order to rationalize the better 2PA cross-sections of **2a**,**b** relative to their fluorenyl homologues **3a**,**b'**, the structural factors rather than the conformational issues must be considered, since both families of compounds are fully planar. First, regardless of the actual nature of the terminal substituents (compare Scheme 8a,b), we can state that the lower aromaticity of the cyclopentadithienyl unit allows a better delocalization of charge between the terminal substituents and the central core compared to the fluorenyl unit, a feature which will also be beneficial to 2PA, as explained above. This will result in a larger photoinduced charge transfer (CT) in the peripheral branches for **2a**,**b** (Table 5) and boost the 2PA cross-section of the corresponding quadrupoles. In addition, the sulfur atoms in **2a**,**b** will also likely contribute to an increase in 2PA relative to **3a**,**b** because the DFT computations reveal that these polarizable atoms strongly contribute to the LUMO of **1a**,**b** and **2a**,**b** (Figures 5 and 6). Indeed, considering that the lowest energy transitions have a dominant HOMO-LUMO character, these atoms should increase the (hyper)polarizability of the whole $\pi$-manifold [58]. Finally, regarding the magnitude of the 2PA cross-section, the comparison of **2a**,**b** with dithienothiophene (DTT) derivatives such as **17a-c** is of interest since these species have a similar number of effective electrons ($N_{eff}$) for a given set of endgroups [10]. The comparison reveals that **2a**,**b** are better two-photon absorbers than **17a-c**, as the 2PA cross-sections of the latter are closer to those of the non-rigid bithiophenes **1a**,**b** [59]. The fact that that the cyclopentadithiophenes **2a**,**b** are also much more soluble in common organic solvents than the DTT derivatives **17a**,**b** makes the former compounds much more attractive for bio-related applications.

**Scheme 8.** Valence-bond mesomeric forms in (pseudo-)quadrupoles **1a,b/2a,b** (**a**) and **3a,b** quadrupoles (**b**) with relevant (transition) dipole moment shown. The weight of the polar forms $B_{1-2}$ and $C_{1-2}$ is respectively larger for **2a,b** than for **1a,b** due to planarization of the $\pi$-manifold.

*Oxygen Photosensitization.* All of the new compounds also photosensitize oxygen. Compared to the 2,7-fluorene homologues **3a,b′**, which should be far less active because of their large fluorescence quantum yield (Table 1), photosensitization can be seen as another positive effect resulting from the presence of sulfur atoms in **1a,b** and **2a,b**. As mentioned above, the sulfur atoms induce more efficient intersystem crossing, leading to an increased population of the triplet excited state at the lowest energy ($T_1$) after photoexcitation [49]. Increased intersystem crossing is experimentally suggested by the shorter lifetimes found for **1a,b** and **2a,b** compared to **3a,b** (Table 2), which roughly inversely correlate with the oxygen photosensitization quantum yields of these compounds (Table 1). Furthermore, the theoretical fluorescence lifetime ($\tau_R$) computed for **1a,b** and **2a,b** (i.e., the lifetime in the absence of any non-radiative deactivation, such as internal conversion and/or isc) is fairly constant for **1a,b** and **2a,b** (1.5–1.8 ns) and only slightly smaller than that found for **3a** and **3b′** at ambient temperatures in a solution (2.1–1.8 ns). This clearly indicates that the shorter global lifetime ($\tau$) experimentally found for **1a,b** and **2a,b** in their first singlet state is due to the increased participation of non-radiative deactivation processes (see $k_{NR}$ in Table 2), which most likely corresponds to intersystem crossing. This partly explains their increased reactivities toward oxygen compared to **3a** and **3b′**. However, given that the spin density computed on the central bithiophene unit changes only modestly between the $T_1$ states, and that is seemingly decorrelated from their oxygen photosensitization quantum yields (Table 1), and also given that the redox potentials of **1a,b** and **2a,b** might favor the formation of charge-transfer complexes with oxygen, any definitive interpretation of the actual reasons for the improved photosensitizing ability of the bithiophenes **1a,b** compared to cyclopentadienyls **2a,b** is not possible at present. It is, however, important to point out that the $T_1$ energies calculated for **1a,b** and **2a,b** (Figure 8) are all lower than those previously reported at the same level of theory for **3a,b** ($\geq 1.18$ eV for **3a**); they are now closer to the first singlet states of oxygen [28], perhaps also explaining the greater photosensitizing ability of the former compounds. Furthermore, the latter compounds have oxidation potentials closer to 1.9 V vs. SCE, a threshold that renders the formation of charge-transfer complexes unlikely [47]. For both families of thiophene-containing compounds, the sum of the fluorescence quantum yield and the quantum yield of oxygen photosensitization

is nearly quantitative, indicating that most of the initially photogenerated singlet states decays by one of these processes [21]. With quantum yields for internal conversion lower than 22% (value for **2b**), any "unproductive" processes for bio-oriented applications will be minimal for these compounds (Table 1).

*Potential Bio-Oriented Uses.* The Stokes shifts of **1a,b** and **2a,b** are sufficient to avoid any detrimental self-absorption, therefore these compounds may have potential for fluorescence imaging purposes. Based on their brightness ($\varepsilon_{max}\ \phi_F$) [60], the cyclopentadithiophene derivatives **2a,b** appear more promising than their bithiophene homologues **1a,b** for 1PA fluorescence imaging, but both of these families of compounds are less efficient one-photon absorbers than their fluorene analogues **3a** or **3b'**. **1a,b** and **2a,b** exhibit more efficient molecular weight-scaled 2PA cross-sections than **3a,b**, and have possibilities in two-photon fluorescence imaging or photodynamic therapy (PDT). Finally, regarding oxygen photosensitization, although better yields were obtained with one-photon excitation for **1a,b**, the $\sigma_2\ \phi_\Delta$ figures of merit reveal much better performances for **2a–b** with two-photon excitation. To date, only charged (water-soluble) bithiophene derivatives such as **18** were successfully used for bioimaging by Marder and coworkers [19–21], but we are not aware of the use of any such derivatives for PDT, although the capability of a water-soluble bithiophene derivative to photosensitize oxygen was very recently reported by the same authors [21]. The few experiments previously attempted with model bithiophene derivatives such as **15** (Scheme 7) were indicative of solubility problems in physiological media [18]. Given that the compounds from the present study are not soluble in aqueous media, prior *ad hoc* functionalization will be required for bio-oriented applications.

## 4. Materials and Methods

### 4.1. General

All the manipulations were carried out under an inert atmosphere of argon with dried and freshly distilled solvents [61]. The transmittance-FTIR spectra were recorded using a Perkin Elmer Spectrum 100 spectrometer (Perkin Elmer, Waltham, MA, USA) equipped with a universal ATR sampling accessory (400–4000 cm$^{-1}$). Nuclear magnetic resonance spectroscopy was performed using a Bruker AV-300 (300 MHz for $^1$H, 75 MHz for $^{13}$C, 228 MHz for $^{19}$F) spectrometer at ambient temperature. The $^1$H and $^{13}$C spectra were calibrated using residual solvent peaks [62,63]. The HRMS analyses were performed at the "Centre Regional de Mesures Physiques de l'Ouest" (CRMPO, Université de Rennes, Rennes, France) on high-resolution Bruker Maxis 4G (Karlsruhe, Germany) or Thermo Fisher Q-Extractive Spectrometers (Waltham, MA, USA). The reactions were monitored by thin-layer chromatography on Merck (Darmstadt, Germany) silica gel 60 F254 precoated aluminum sheets or by NMR spectroscopy. The chromatographic separations (column chromatography or flash chromatography) were performed on Merck silica gel (40–63 μm), Aldrich basic alumina (activity 1) or Aldrich neutral alumina (activity 1) with the eluants indicated (Merck, Darmstadt, Germany). The commercial reagents and (pre-/co-)catalysts were used as received. *N,N*-diphenyl-4-[2-(trimethylsilyl)ethynyl]-aniline (**5**) [29], 5,5'-diethynyl-2,2'-bithiophene (**6b**) [30], 1-bromo-4-[(trifluoromethyl)sulfonyl]benzene (**7**) [25], 4,4-dibutyl-4*H*-cyclopenta [2,1-*b*:3,4-*b'*]dithiophene (**8**) [32], *N,N*-diphenyl-4-ethynylaniline (**10**) [33] and 1-[(trifluoromethyl)sulfonyl]-4-[2-(trimethylsilyl)ethynyl]benzene (**11**) [25] were synthesized as described in the literature.

### 4.2. Synthetic Procedures

*4,4'-([2,2'-bithiophene]-5,5'-diylbis(ethyne-2,1-diyl))bis(N,N-diphenylaniline)* (**1a**). To a mixture of 5,5'-dibromo-2,2'-bithiophene (**4**; 78.3 mg, 0.242 mmol), *N,N*-diphenyl-4-[2-(trimethylsilyl)ethynyl]benzenamine (**5**; 198.3 mg, 0.581 mmol), Pd(PPh$_3$)$_2$Cl$_2$ (9.3 mg, 0.0132 mmol, 5.5%) and CuI (2.2 mg, 0.0115 mmol, 5%) in toluene/NEt$_3$ [5:1] mixtures (1.2 mL) under argon atmosphere, TBAF (1 M in THF, 0.250 mL, 0.250 mmol) was added. After stirring at 85 °C for 36 h, the solvents were evaporated under reduced pressure and the residue was purified by column chromatography (heptane/CH$_2$Cl$_2$ 90:10 then

65:35 eluents), leading to isolation of 55.4 mg (33%) of **1a** as an orange solid. $R_f$: 0.18 (4:1 heptane/$CH_2Cl_2$). MP 235 °C. HRMS (ESI): $m/z$ = 700.2005 $[M]^+$ (calcd for $C_{48}H_{32}N_2S_2$: 700.20014). $^1H$ NMR (300 MHz, CDCl$_3$): $\delta$ = 7.37 (d, $J$ = 8.7 Hz, 4H), 7.33–7.28 (m, 8H), 7.16–7.07 (m, 16H), 7.02 (d, $J$ = 8.7 Hz, 4H). $^{13}C\{^1H\}$ NMR (75 MHz, CDCl$_3$, ppm): $\delta$ = 148.3, 147.2, 137.9, 132.5, 132.4, 129.6, 125.3, 124.0, 123.8, 123.1, 122.2, 115.4, 95.1, 81.9. IR (cm$^{-1}$): 2181 (w, C≡C). Raman (cm$^{-1}$): 2185 (w, C≡C).

*5,5′-bis((4-((trifluoromethyl)sulfonyl)phenyl)ethynyl)-2,2′-bithiophene* (**1b**). A mixture of 5,5′-diethynyl-2,2′-bithiophene (**6b**; 70.4 mg, 0.329 mmol), 1-bromo-4-[(trifluoro-methyl) sulfonyl]benzene (**7**; 290.0 mg, 1.00 mmol), Pd(PPh$_3$)$_2$Cl$_2$ (24.3 mg, 0.0346 mmol, 10%) and CuI (7.6 mg, 0.04 mmol, 12%) in DMF/NEt$_3$ [6:1] (3.5 mL) was stirred at 80 °C for 16 h under an argon atmosphere. $CH_2Cl_2$ was added, and the organic layer was washed with water (4×). The solvents were evaporated under reduced pressure and the residue was purified by two successive chromatographic separations (1st column: $CH_2Cl_2$; 2nd column: $CCl_4$/$CH_2Cl_2$ [90:10] eluents), leading to isolation of 62.5 mg (30%) of **1b** as a pale-orange solid. $R_f$: 0.37 (1:1 heptane/$CH_2Cl_2$). MP 300 °C. HRMS (ESI): $m/z$ = 652.9419 $[M + Na]^+$ (calcd for $C_{26}H_{12}O_4F_6NaS_4$: 652.9415). $^1H$ NMR (300 MHz, CDCl$_3$): $\delta$ = 8.05 (d, $J$ = 8.5 Hz, 4H), 7.79 (d, $J$ = 8.5 Hz, 4H), 7.34 (d, $J$ = 3.9 Hz, 2H), 7.20 (d, $J$ = 3.9 Hz, 2H). $^{19}F\{^1H\}$ NMR (282 MHz, CDCl$_3$, ppm): $\delta$ = −78.2 ppm. IR (cm$^{-1}$): 2194 (m, C≡C). Raman (cm$^{-1}$): 2186 (w, C≡C).

*4,4′-((4,4-dibutyl-4H-cyclopenta[2,1-b:3,4-b′]dithiophene-2,6-diyl)bis(ethyne-2,1-diyl))-bis (N,N-diphenylaniline)* (**2a**). A mixture of 4,4-dibutyl-2,6-diiodo-*4H*-cyclopenta [2,1-b:3,4-b′]dithiophene (**9**; 50.8 mg, 0.0937 mmol), *N,N*-diphenyl-4-ethynylaniline (**10**; 54.8 mg, 0.203 mmol), Pd(PPh$_3$)$_2$Cl$_2$ (4.1 mg, 0.0058 mmol, 6%) and CuI (1.7 mg, 0.0089 mmol, 9%) in NEt$_3$ (1 mL) was stirred at 45 °C for 16 h under an argon atmosphere. The reaction mixture was filtered through a Celite pad with $CH_2Cl_2$ as a washing solvent. After evaporation of the solvents, the residue was purified by column chromatography with eluents heptane/$CH_2Cl_2$ [9:1] mixtures, then [8:2] mixtures, leading to isolation of 34.9 mg (45%) of **2a** as a brown solid. $R_f$: 0.44 (3:2 heptane/$CH_2Cl_2$). MP 170–180 °C (dec.). HRMS (MALDI, DCTB): $m/z$ = 824.3230 $[M]^+$ (calcd for $C_{57}H_{48}N_2S_2$: 824.32534). $^1H$ NMR (300 MHz, CDCl$_3$): $\delta$ = 7.35 (d, $J$ = 8.7 Hz, 4H), 7.34–7.22 (m, 8H), 7.15–7.03 (m, 14H), 7.02 (d, $J$ = 8.7 Hz, 4H), 1.89–1.75 (m, 4H), 1.26–1.08 (m, 4H), 1.01–0.87 (m, 4H), 0.78 (t, $J$ = 7.3 Hz, 6H). $^{13}C\{^1H\}$ NMR (75 MHz, CDCl$_3$, ppm): $\delta$ = 158.5, 148.1, 147.3, 137.8, 132.4, 129.6, 126.1, 125.2, 123.9, 123.8, 122.3, 115.8, 95.0, 83.4, 54.1, 37.8, 26.8, 23.2, 14.0. IR (cm$^{-1}$): 2184 (vw, C≡C).

*4,4-dibutyl-2,6-bis((4-((trifluoromethyl)sulfonyl)phenyl)ethynyl)-4H-cyclopenta[2,1-b:3, 4-b′] dithiophene-sulfur dioxide (1/1)* (**2b**). To a mixture of 4,4-dibutyl-2,6-diiodo-*4H*-cyclopenta [2,1-b:3,4-b′]dithiophene (**9**; 50.1 mg, 0.0924 mmol), 1-[(trifluoromethyl)sulfonyl]-4-[2-(trimethyl-silyl)ethynyl]benzene (**11**; 72.0 mg, 0.235 mmol), Pd(PPh$_3$)$_2$Cl$_2$ (4.0 mg, 0.0057 mmol, 6%) and CuI (1.3 mg, 0.0068 mmol, 7%) in NEt$_3$ (1 mL) under an argon atmosphere, TBAF (1 M in THF, 0.230 mL, 0.230 mmol) was added. After stirring at 45 °C for 16 h, the reaction mixture was filtered through a Celite pad with $CH_2Cl_2$ as a washing solvent. The solvents were evaporated under reduced pressure and the residue was purified by column chromatography with a heptane/$CH_2Cl_2$ [7:3] eluent, leading to isolation of 34.9 mg (50%) of **2b** as an orange solid. $R_f$: 0.19 (6:4 heptane/methylene chloride). MP 226 °C. HRMS (MALDI, DCTB): $m/z$ = 754.0780 $[M]^+$ (calcd for $C_{35}H_{28}O_4F_6S_4$: 754.07692). $^1H$ NMR (300 MHz, CDCl$_3$): $\delta$ = 8.03 (d, $J$ = 8.5 Hz, 4H), 7.76 (d, $J$ = 8.5 Hz, 4H), 1.90 (m, 4H), 1.22 (m, 4H), 0.97 (m, 4H), 0.82 (t, $J$ = 7.3 Hz, 6H). $^{13}C\{^1H\}$ NMR (75 MHz, CDCl$_3$, ppm): $\delta$ = 159.7, 139.8, 132.0, 130.9, 129.9, 128.15, 122.7, 119.9 (q, $J$ = 326.0 Hz), 93.2, 90.7, 54.4, 37.7, 26.8, 23.1, 14.0. $^{19}F\{^1H\}$ NMR (282 MHz, CDCl$_3$, ppm): $\delta$ = −78.2 ppm. IR (cm$^{-1}$): 2186 (m, C≡C). Raman (cm$^{-1}$): 2186 (vw, C≡C).

*2,6-Iodo-4,4-dibutyl-4H-cyclopenta [2,1-b:3,4-b′]dithiophene* (**9**). *N*-Iodosuccinim-ide (NIS) (162.7 mg, 0.723 mmol) was added at 0 °C to a solution of 4,4-dibutyl-*4H*-cyclopenta [2,1-b:3,4-b′]dithiophene (**8**; 99.7 mg, 0.343 mmol) in 5 mL of a [4:1] ($v/v$) mixture of THF and acetic acid. The mixture was stirred at 0 °C for 30 min, then allowed to warm to room temperature overnight. The solvents were evaporated, and the residue was purified by

a column chromatography eluent (heptane) to give 166.4 mg (89%) of **9** as a pale-yellow solid (that needs to be stored in the cold). $R_f$: 0.52 (heptane). MP 100 °C. HRMS (ASAP): $m/z$ = 541.9089 $[M]^+$ (calcd for $C_{17}H_{20}I_2S_2$: 541.90905). $^1$H NMR (300 MHz, CDCl$_3$): $\delta$ = 7.10 (s, 2H), 1.77 (m, 4H), 1.17 (m, 4H), 0.90 (m, 4H), 0.80 (t, $J$ = 7.3 Hz, 6H). $^{13}$C{$^1$H} NMR (75 MHz, CDCl$_3$, ppm): $\delta$ = 158.2, 141.0, 130.9, 71.9, 54.0, 37.4, 26.6, 23.0, 13.9.

### 4.3. Electronic Absorption and Emission Measurements

All the photophysical measurements were performed on freshly prepared air-equilibrated toluene solutions (HPLC grade) at room temperature (298 K). The UV–Vis absorption spectra were recorded on dilute solutions (ca. $10^{-5}$ M) using a Jasco V-570 spectrophotometer (Jasco, Mary's Court Easton, MD, USA). Steady state fluorescence studies were performed in dilute air-equilibrated solutions in quartz cells of 1 cm path length (ca. $1 \times 10^{-6}$ M, optical density < 0.1) at room temperature (20 °C) using an Edinburgh Instruments (FLS920) spectrometer (Edinburgh Instruments, Edinburgh, UK) in photon-counting mode. The fully corrected excitation and emission spectra were obtained with an optical density at $\lambda_{exc} \leq 0.1$. The fluorescence quantum yield of each compound was calculated using the integral of the fully corrected emission spectra relative to a standard, quinine bisulfate (QBS, $\lambda_{exc}$ = 346 nm, $\Phi_F$ = 0.546) [64,65]. The UV-Vis absorption spectra used for the calculation of the fluorescence quantum yields were recorded using a double-beam Jasco V-570 spectrometer.

### 4.4. Two-Photon Absorption Experiments

To span the 790–920 nm range, a Nd:YLF-pumped Ti:sapphire oscillator (Chameleon Ultra, Coherent, Santa Clara, CA, USA) was used generating 140 fs pulses at an 80 MHz repetition rate. The excitation power was controlled using neutral density filters of varying optical density mounted in a computer-controlled filter wheel. After five-fold expansion through two achromatic doublets, the laser beam was focused by a microscope objective (10×, NA 0.25, Olympus, Olympus, Shinjuku, Tokyo, Japan) into a standard 1 cm absorption cuvette containing the sample. The applied average laser power arriving at the sample was typically between 0.5 and 40 mW, leading to a time-averaged light flux in the focal volume on the order of 0.1–10 mW/mm$^2$. The fluorescence intensity was measured at several excitation powers in this range by employing the filter wheel. For each sample and each wavelength, the quadratic dependence of the fluorescence intensity (F) on the excitation intensity (P), i.e., the linear dependence of F on P$^2$ was systematically checked (see ESI, Figure S10). The fluorescence from the sample was collected in epifluorescence mode, through the microscope objective, and reflected by a dichroic mirror (Chroma Technology Corporation, Bellows Falls, VT, USA; "red" filter set: 780dxcrr). This made it possible to avoid the inner filter effects related to the high dye concentrations used ($10^{-4}$ M) by focusing the laser near the cuvette window. The residual excitation light was removed using a barrier filter (Chroma Technology; "red": e750sp–2p). The fluorescence was coupled into a 600 µm multimode fiber by an achromatic doublet. The fiber was connected to a compact CCD-based spectrometer (BTC112-E, B&W Tek, Plainsboro Township, NJ, USA), which measured the two-photon excited emission spectra. The emission spectra were corrected for the wavelength-dependence of the detection efficiency using correction factors established through the measurement of reference compounds having known fluorescence emission spectra. Briefly, the set-up allowed for the recording of the corrected fluorescence emission spectra under multiphoton excitation at variable excitation power and wavelength. The 2PA cross-sections ($\sigma_2$) were determined from the two-photon excited fluorescence (TPEF) action cross-sections ($\sigma_2 \Phi_F$) and the fluorescence emission quantum yield ($\Phi_F$). The TPEF cross-sections of $10^{-4}$ M toluene solutions were measured relative to fluorescein in 0.01 M aqueous NaOH using the well-established method described by Xu and Webb [66] and the appropriate solvent-related refractive index corrections [67]. To verify the absence of aggregation, the UV-visible absorption spectra of the dyes were

recorded at this concentration in cells of 1 mm pathlength and compared with those obtained with the diluted solutions in cells of 1 cm pathlength.

### 4.5. Computational Details

The ground state (GS) structures ($S_0$ state) were optimized using the PBE0 functional [27] with the 6-31 + G(d) basis set and verified to be true minima on their potential energy surface (PES). All the calculations were carried out taking into account the solvation (toluene solvent) with the polarizable continuum model (PCM) [68–71]. All the optical properties (absorption and emission) were simulated using the TD-DFT approach and using not only the PBE0 functional, but also the CAM-B3LYP functional [72] to account for the long-distance electronic transfers that may occur in such compounds upon excitation. The vibration frequency calculations were also performed on the $S_1$ excited state geometries after TD-DFT optimization. In order to reach the highest accuracy with the emission spectra, the vibronic effects were taken into account using the Adiabatic Hessian (AH) model [73–79]. This model, which explicitly includes mode mixing between fundamental and excited states, accurately reproduces the optical properties of molecular systems. To ensure a sufficient spectrum progression (>90%), normal modes with the lowest energies were neglected in the vibronic treatment. The optimized geometries and vibration frequencies of the first triplet state ($T_1$) were obtained using spin-unrestricted calculations. All the computations were carried out using the *Gaussian* 09 [80] and *Gaussian 16* programs [81]. Molecular structures, molecular orbitals, UV-visible and emission spectra, as well as the electron density differences maps between the respective vertical excited states $S_1$ and the ground states $S_0$ were also plotted. The 3D charge density and electron spin density maps were drawn using the *VESTA* software (Version 3.5.7, K. Momma, F. Izumi, Tsukuba, Ibaraki 305-0044, Japan) [82].

The 2PA cross-sections were calculated using the damped cubic response theory module of Jensen et al. [83] implemented in the Amsterdam Density Functional (*ADF2018* and *ADF2019*) program packages [84,85]. The non-linear response properties were calculated using the statistical average of orbital model exchange-correlation potentials (SAOP) model. The SAOP potential model [86,87] was chosen due to its correct Coulombic decay of the potential at long distances, which is important for the description of response properties. Moreover, a DZP basis set was used to limit the computational requirements, and the solvent effects were not included in these simulations. No symmetry constraint was applied. In previous work, we demonstrated that SAOP/DZP calculations lead to 2PA cross-sections in satisfying agreement with experimental measurements [23,50]. The excited-state lifetime was included in the theory using a damping parameter $\Gamma = 0.0034$ a.u. (~0.1 eV~800 cm$^{-1}$) values [88], which was previously found to be acceptable for 2PA computations [83,89]. The SAOP/DZP MO surfaces were visualized with the *ADFView* program [90]. As is generally accepted, in order to simulate the 2PA spectra, the calculations were carried out point by point for each laser energy. The imaginary part ($\gamma_{im}$) of the averaged third-order hyperpolarizability $\gamma$ permits expression of the 2PA cross-section $\sigma_2$ as: [83,91,92]

$$\sigma_2(\omega) = \frac{N\pi^3 \alpha_f^2 \hbar^3 \omega^2}{15e^4} \gamma_{im} \tag{2}$$

with:

$$\gamma_{im} = \sum_{\alpha\beta} Im\left[\gamma_{\alpha\alpha\beta\beta}(-\omega;\omega,\omega',-\omega) + \gamma_{\alpha\beta\beta\alpha}(-\omega;\omega,\omega',-\omega) + \gamma_{\alpha\beta\alpha\beta}(-\omega;\omega,\omega',-\omega)\right] \tag{3}$$

where $\alpha_f$ is the fine structure constant, $\omega$ is the photon energy, $\hbar$ is the reduced Planck's constant and $e$ is the elementary charge. The integer value $N$ is related to the experimental setup, and in this work, $N = 4$ was used for all the simulated 2PA spectra [83,91]. The cross-section $\sigma_2$ unit are Goeppert-Mayer (1 GM = $10^{-50}$ cm$^4$ s photon$^{-1}$) [93]. The $\sigma^{2PA}$ values were first obtained in atomic units (a.u.) and then converted with ($0.529177 \times 10^{-8}$ cm/a.u.)$^4$

$\times$ (2.418884 $\times$ 10$^{-17}$ s/a.u.) to give the conventional units (cm$^4$ s photon$^{-1}$). The corresponding expression for $\Upsilon^{2PA}$ shown in Equation (3) avoids any negative 2PA intensities caused by the pure one-photon processes and allows for appropriate $\sigma_2$ calculations in the presence of one- and two-photon double-resonance effects [94,95]. We therefore adopted it for all the 2PA simulations in this work.

## 5. Conclusions

New (quasi-)quadrupolar bithiophenes (**1a,b**) and cyclopentadithiophenes (**2a,b**) bearing either electron-donating (NPh$_2$) or electron-withdrawing (SO$_2$CF$_3$) endgroups at their periphery have been synthesized, characterized and subsequently modelled by DFT. While we showed computationally that the presence of various conformers is likely to occur in a solution at ambient temperature for these compounds, we also demonstrated that the most stable species among them can be confidently used to model the experimental photophysical properties in a solution. All of these derivatives exhibit good light harvesting efficiency due to their strong absorption bands in the visible range. A fair match with the experiment was obtained by TD-DFT using the CAM-B3LYP functional and we demonstrated that this strongly allowed absorption corresponds to the HOMO-LUMO transition in **1a,b** or **2a,b**. This populates a $\pi$-$\pi$* excited state (S$_1$) via a transition with a CT character between the branches and the core, directed from the electron-rich part to the electron-poor part, and thus depending on the nature of the peripheral substituents. Using a SAOP model, we next verified that, for all of these quadrupolar derivatives, regardless of the existence of actual centrosymmetry, the exclusion rule is roughly obeyed, so that the first intense 2PA absorption takes place into the second singlet state S$_2$, a $\pi$-$\pi$* excited state related to S$_1$, but of different symmetry (*g* vs. *u*). These compounds can therefore be efficiently excited by two-photon absorption. As demonstrated by figures of merit, the cyclopentadithiophenes **2a,b** are significantly better two-photon absorbers, which we attribute to the enforced planarity of these derivatives compared to **1a,b**. From an applied perspective, **2a,b** appear more soluble in common organic solvents than their bithiophene homologues **1a,b**. Finally, we have experimentally shown that all new derivatives can photosensitize molecular oxygen and that better quantum yields are obtained with bithiophenes (**1a,b**) than with cyclopentadithiophenes (**2a,b**) upon one-photon excitation. While this trend is difficult to rationalize in the light of the available data, it might be (at least in part) correlated with the efficiency of intersystem crossing to the triplet state.

When the photophysical properties of **2a,b** were compared to those of their known fluorene homologues (**3a,b**), it appears that replacement of the 1,4-phenylene by 2,5-thienyl units generates new fluorescent dyes more suitable for biophotonic uses. While this structural change favors electron delocalization between the peripheral positions, we also demonstrated via DFT results that the insertion of sulfur in the carbon framework increases the electron-richness and hyperpolarizability of the central polycyclic aromatic unit, decreasing the overall bandgap of the compound as well as its oxidation potential. The sulfur atoms also promote intersystem crossing. All of these electronic changes proceeding from **3a,b** to **2a,b** contribute to (i) a red-shift in their 2PA maxima, (ii) a boost in their 2PA cross-sections and (iii) an increased population of triplet excited states (subsequent to excitation), which enhances the capability to photosensitize oxygen with only minor losses by internal conversion. These structural changes therefore significantly optimize the photophysical properties of interest for bio-oriented applications. In light of the recent upsurge in interest for water-soluble bithiophene derivatives in two-photon bioimaging due to their versatility for staining specific cellular compartments, the present studies suggest a role for cyclobithiophene analogues for similar uses. The present contribution strongly suggests that water-soluble derivatives of **2a,b** would be promising new dyes for two-photon photodynamic therapy (PDT).

**Supplementary Materials:** The following supporting information can be downloaded at: https://www.mdpi.com/article/10.3390/photochem3010009/s1, Figures S1–S7: $^1$H/$^{13}$C{$^1$H}/$^{19}$F{$^1$H} NMR spectra for the new compounds **1a,b**, **2a,b** and **9**, Figures S8–S9: Two-photon excited fluorescence (2PEF) data for **1a,b** and **2a,b**, Figures S10–S14: Conformers and conformational freedom at the PBE0/6-31+G(d) level of theory for **1a,b** and **2a,b**, Figure S15: Vibronic calculations for emissions of **1a,b** and **2a,b**, Figure S16: Calculated spin density in the first triplet (T$_1$) excited state of **1a,b** and **2a,b**, Figure S17: Calculated one-photon and two-photon absorption for **1a,b** and **2a,b** using SOAP functionals; Tables S1 and S2: Conformers and conformational freedom at the PBE0/6-31+G(d) level of theory for **1a,b** and **2a,b**, Tables S3–S6: TD-DFT calculations for **1a,b** and **2a,b**, Tables S7–S14: Calculated one-photon and two-photon absorption for **1a,b** and **2a,b** using SOAP functionals, Tables S15–17. Optimized Cartesian coordinates for **1a,b** and **2a,b** using PBE1PBE/6-31+G(d) in toluene; Scheme S1: Conformers and conformational freedom at the PBE0/6-31+G(d) level of theory for **1a,b** and **2a,b**.

**Author Contributions:** Conceptualization, F.P., J.-F.H. and A.B.; Funding acquisition, F.P., A.B. and M.G.H.; Investigation, N.R., S.G. and A.T.; Methodology, C.L. and N.R.; Resources, M.B.-D.; Supervision, A.B., S.M., J.-F.H., O.M., F.P. and M.G.H.; Writing—original draft, S.G. and O.M.; Writing—review & editing, F.P., J.-F.H., A.B., M.G.H. and O.M. All authors have read and agreed to the published version of the manuscript. The two authors N.R. and S.G. contributed equally.

**Funding:** This project was supported by Région Bretagne (ARED Project N° 8394 *Bifocrom*, PhD grant for A.T.), the CNRS (*REDOCHROM* LIA & *MAITAI* IRP Projects), GENCI-IDRIS and GENCI-CINES (Grant No. 2021-080649).

**Data Availability Statement:** Data is contained within the article or Supplementary Material.

**Acknowledgments:** The Region Bretagne and CNRS are acknowledged for financial support. The authors are grateful to the French GENCI-IDRIS and GENCI-CINES for an allocation of computing time.

**Conflicts of Interest:** The authors declare no conflict of interest.

**Sample Availability:** Samples of the compounds are available from the authors.

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
