# Peer review of "Linear and Nonlinear Optical Properties of Quadrupolar Bithiophenes and Cyclopentadithiophenes as Fluorescent Oxygen Photosensitizers"

_2673-7256, doi:10.3390/photochem3010009_

Round 1

Reviewer 1 Report

After carefully reading the manuscript entitled “Linear and Nonlinear Optical Properties of Quadrupolar Bithiophenes and Cyclopentadithiophenes as Fluorescent Oxygen Photosensitizers” by Nicolas Richy and co-workers, I strongly recommend the publication in Photochem as it is.

The manuscript shows very interesting experimental linear and nonlinear spectroscopic results for a new synthesized quadrupolar thiophenes derivatives. The results are in agreement with similar molecular structures with similar effective electron numbers. Concerning the photophysical properties, the authors have shown complete data on linear optical properties. Also, the two-photon absorption spectra were obtained through excited fluorescence measurements, which were well performed, and given a good spectral resolution. The results were explained in terms of electron-withdrawing groups at the periphery in combination with the increased electron-richness of the central core. Also, quantum chemistry calculations were performed to support the experimental data.

According to the authors, the region with the highest 2PA cross-sections is concerned with an S2 state, which is 1PA forbidden. This feature, concerning the molecular structure symmetry, is also observed in distinct chromophores such as oligofluorenes.

Another important point in the manuscript is the singlet oxygen generation, which depicts relatively high quantum yields obtained by 1PA and analyzed as well by 2PA excitations. The interesting authors' observation is that cyclopentabithiophenes are more efficient bithiophenes when 2PA is used, which brings an interesting application to the first biological window. The results were also reinforced by quantum chemistry calculation, showing triplet states energy.

Author Response

We thank the Reviewer for his rewarding comments.

Reviewer 2 Report

In this manuscript, Richy et. al. studied the optical properties of a series of bithiophenes and cyclopentadithiophenes derivatives by an experimental and computational approach proposing these compounds as efficient quadrupolar fluorescent molecules for oxygen photosensitizers. The molecules were satisfactorily obtained and characterized by MS and NMR methods and their linear and nonlinear optical properties obtained by experimental and DFT methods. This work is well performed and discussed, although there are some drawbacks that must be previously clarified.

1.- Although the authors mentioned that the bithiophenes and cyclopentadithiophenes derivatives can be considered as quadrupolar systems, Regarding symmetry considerations, compounds 2a,b can also be labeled as C2v compounds. Did you considered measuring/evaluating the dipole moment to confirm this statement?

2.- Considering the previous comment, and also the non-centrosymmetric 2a,b compounds, these compounds present a low energy transitions in which the HOMO-LUMO is expected to be symmetry-forbidden. How do you explain that in your analysis (table 3) the H-L contribution is larger in compounds 2a,b, which are “more C2v” than the 1a,b compounds?

3.- Please clarify the asseveration of “significant Charge Transfer”, while in Fig 6, the HOMO – LUMO composition, specifically for compounds 1b and 2b the electronic distribution is not showing this effect.

4.- As mentioned above, despite the work is well done and presented, the reading is sometimes difficult or hard to understand for a common reader. I recommend that the “discussion” section be included in results to then give the conclusions. Also, I would recommend moving section 4 “materials and methods” after the conclusions, or after introduction, in order to improve the reading process from results to conclusions

5.- In scheme 2: please, include the missing conditions for the compound 6.

6.-In the computational methods, section 4.5, there is not clear how the authors obtained the excited states to evaluate the emission properties of the studied compounds. Please clarify

Minor issues:

7.- In Table 3, there are some “*” with without explanation in the corresponding caption. In the same table 3 there are also some misaligned data.

8.- Apparently in Table 1, the absorption wavelength for compound 1a is incorrect. 30 nm is too far from the UV range

9.- there are some changes in the fonts/styles in the whole manuscript. Check please.

Author Response

CHANGES REQUESTED BY REVIEWER 2

In this manuscript, Richy et al. studied the optical properties of a series of bithiophenes and cyclopentadithiophenes derivatives by an experimental and computational approach proposing these compounds as efficient quadrupolar fluorescent molecules for oxygen photosensitizers. The molecules were satisfactorily obtained and characterized by MS and NMR methods and their linear and nonlinear optical properties obtained by experimental and DFT methods. This work is well performed and discussed, although there are some drawbacks that must be previously clarified.

1) Although the authors mentioned that the bithiophenes and cyclopentadithiophenes derivatives can be considered as quadrupolar systems, Regarding symmetry considerations, compounds 2a,b can also be labeled as C2v compounds. Did you considered measuring/evaluating the dipole moment to confirm this statement?

Referee is right regarding the C2v symmetry of the 2a,b derivatives; accordingly, we have now precised that point by adding « (C2v symmetry with ideal conformations) » in text p. 7 (section 2.2. ; Two-Photon Absorption) and p. 18 (section 3 ; Conformers and Symmetry).  However, in the optimized GS conformations the terminal groups adopt orientations avoiding a higher symmetry group than Cs. This has also now been precised p. 8 (section 2.3.; Ground State Geometry) by adding the sentence: «All belong to the Cs symmetry group and none of them presents a higher symmetry. » Due to that observation and to the fact that the compound most likely equilibrate in solution with other conformers at higher energy, we did not attempt to experimentally determine their dipole moments which were likely weak for 2a and 2b and not depending from a single conformation. But the referee is right that they should be likely higher for 2a and 2b than for 1a and 1b due to these symmetry issues.

2) Considering the previous comment, and also the non-centrosymmetric 2a,b compounds, these compounds present a low energy transitions in which the HOMO-LUMO is expected to be symmetry-forbidden. How do you explain that in your analysis (table 3) the H-L contribution is larger in compounds 2a,b, which are “more C2v” than the 1a,b compounds?

Actually, the HOMO-LUMO (S1) transitions are always highly allowed for all these systems, as indicated by their high oscillator strengths values. The first forbidden (S2) transitions are either HOMO-LUMO+1 transitions or HOMO-1-LUMO transition (see Table 3). As thoughfully observed by the referee, they are always higher (in computations) for compounds 2a,b compared to 1a,b. While this might arise from a more pronounced « non-centrosymmetric » character of the former compounds which helps relaxing the parity rule, due to the fact that all the GS conformers used for TD-DFT computations adopts a Cs geometry, we prefered to aviod commenting on that in order to avoid over-interpretation of the results. We note nevertheless that this this trend seems also followed by experimental observations (Figure 2), but in a less marked way than in computations. These were given only for the most stable GS conformer (a) in Table 3, whereas in solution several different conformers possibly equilibrate in solution (see above).

3) Please clarify the observation of “significant Charge Transfer”, while in Fig 6, the HOMO – LUMO composition, specifically for compounds 1b and 2b the electronic distribution is not showing this effect.

The referee is right in the fact that the net charge transfer in S1 is not so large (around 0.5 e) for all compounds (and not only 2a and 2b which MOs are given in Fig. 6), as illustrated in Fig. 7. By significant, we intended « non-negligible », but may be this adjective was confusing, so we have now replaced it by « some » on p. 4 (section 2.2.; Photophysical Properties)

4) As mentioned above, despite the work is well done and presented, the reading is sometimes difficult or hard to understand for a common reader. I recommend that the “discussion” section be included in results to then give the conclusions. Also, I would recommend moving section 4 “materials and methods” after the conclusions, or after introduction, in order to improve the reading process from results to conclusions.

Although it apparently reads less easily according to this referee (referee 1 did not comment about that), we prefer to maintain separated the « Results » and « Discussion » sections because it allows us to cumulate evidences from different sources during our discussion, which would not have been so easy with a fused « Result and Discussion » section. By these means, we also avoid unnecessary repetition of arguments and can put more easily the present results in perspective with previous works in the literature. The place of the « Materials and Methods » before the « Conclusion » was imposed by the format of this MDPI journal.

5) In scheme 2: please, include the missing conditions for the compound 6.

These have now been added as b) and c) in the caption of Scheme 2 (p. 3). Note that the compounds 6a,b are known compounds and their synthesis of the latter is referenced at the end of the section 4.1. (General Conditions ) in Materails and Methods (section 4.).

6) In the computational methods, section 4.5, there is not clear how the authors obtained the excited states to evaluate the emission properties of the studied compounds. Please clarify.

It is stated in the Computational Details section (4.5.) that « Vibration frequency calculations were also performed on the S1 excited state geometries after TD-DFT optimization ». The geometry optimization of the S1 excited state is totally automated in the Gaussian programs. The used keywords are “opt TD (full, root=1,nstates=10)”. Then, the vibration frequencies have been computed using the previously optimized geometry, the keyword “freq” replacing “opt”.  Finally the vibronic structure of the fluorescence is obtained with the keywords, “em” for emission and “AH” asking for the desired model, the Adiabatic Hessian one, necessitating as input the normal modes of S0 and S1. For sake of conciseness, we prefer avoid adding these quite technical precisions in the computational section (which seldom appear in manuscripts and with which many theoretical chemists are familiar). We have nevertheless added a sentence to explain how the T1 states were obtained : « The optimized geometries and vibration frequencies of the first triplet state (T1) were obtained using spin-unrestricted calculations.»

Minor issues:

7) In Table 3, there are some “*” with without explanation in the corresponding caption. In the same table 3 there are also some misaligned data.

A footnote has now been added to this Table to clarify the meaning of these asterisks: « * These transitions correspond to the first forbidden 1PA transition which is 2PA allowed. »

8) Apparently in Table 1, the absorption wavelength for compound 1a is incorrect. 30 nm is too far from the UV range.

This typo is now corrected.

9) There are some changes in the fonts/styles in the whole manuscript. Check please.

Done: several sections remaining in Times New Roman style (10 pts) have been found and have been changed for Palatino Linotype (10 pts).